# Combined targeted and epigenetic-based therapy enhances antitumor immunity by stabilizing GATA6-dependent MHCI expression in pancreatic ductal adenocarcinoma

GATA6 promotes epithelial phenotypes and limits epithelial-to-mesenchymal (EMT) transition in pancreatic ductal adenocarcinoma (PDAC). Here we show that GATA6 defines a tumor cell state that induces MHCI expression and anti-tumor cytotoxicity upon therapy. In human PDAC, GATA6 expression correlates with immune cell infiltration, and spatial analysis reveals interaction between GATA6[+] tumor cells and CD8[+] T cells. In murine PDAC, MEK inhibition (MEKi) enriches antigenicity-related gene sets in GATA6[high] cells, while GATA6 knockout or degradation impairs MEKi-induced MHCI upregulation. High-GATA6 tumors respond to MEKi with increased MHCI, enhancing T-cell cyto-toxicity, whereas GATA6 loss abolishes this effect. Treatment-induced EMT reduces GATA6[+] populations and MHCI expression, which is restored by combining MEKi with HDAC inhibitors, enhancing GATA6[+] tumor cells, MHCI, CD8[+] T cell infiltration, tumor suppression, and survival. These findings suggest that therapeutic strategies promoting a GATA6-driven tumor cell state improve immune recognition of PDAC cells and potentiate anti-tumor cyto-toxic effects.

GATA6 is a key regulator of the classical PDAC phenotype, maintaining epithelial differentiation while restraining epithelial-to-mesenchymal transition (EMT)[1]. Low GATA6 expression is linked to the basal-like subtype and poorer survival outcomes[1–5]. Retrospective studies have shown strong associations between high GATA6 expression and pro-longed survival in treatment-naive and advanced PDAC cases[2,6,7]. However, these survival associations are inconsistent following neoadjuvant therapy, likely due to phenotypic plasticity, cell state transitions, and the emergence of intermediary tumor phenotypes during treatment[1,8–11]. Besides subtype classification, GATA6 influences immune surveillance. Its loss in PDAC is associated with impaired antigen processing, reduced CD8[+] T-cell infiltration[12], and increased immunosuppressive myeloid cells in basal-like tumors[13]. Given its pivotal role in shaping tumor phenotype and immune interactions, it is crucial to dissect the dynamic role of GATA6 and its interactions with the TME during therapeutic responses in PDAC.

Frequent downregulation of major histocompatibility complex (MHC) I is a hallmark of solid tumors and is implicated in conferring resistance to immunotherapy[14,15]. Targeted inhibition of MEK (MEKi), a key kinase within the mitogen-activated protein kinase (MAPK) path-way, has emerged as a strategy to modulate tumor immunogenicity and reshape the immune microenvironment[16–20]. However, despite the strong preclinical rationale, the clinical response of PDAC to MAPK pathway inhibition has been disappointing[21,22]. Notably, treatment-

✉e-mail: f.cheung@dkfz-heidelberg.de; j.siveke@dkfz.de

induced phenotypic transitions associated with EMT have been observed, underscoring the adaptive nature of tumor subtypes and highlighting the imperative to disrupt regulatory pathways governing cellular plasticity[23,24], supporting that tumor subtypes are adaptive and denote the importance of perturbing regulatory pathways underlying cellular plasticity. Histone deacetylase inhibitors (HDACi) are known to promote epithelial gene expression in tumor cells and reduce pro-tumorigenic stromal activation in PDAC[25,26], offering a strategy to combat tumor heterogeneity and overcome therapy resistance. HDACi, including Suberoylanilide Hydroxamic Acid (SAHA), moceti-nostat, and domatinostat, have been demonstrated to enhance the expression of antigen-presenting machinery genes and MHC mole-cules, alongside promoting infiltration of cytotoxic T lymphocytes (CTLs), across various solid cancers[27,28].

Here, we investigated GATA6-dependent immunological responses to MEKi using preclinical PDAC models with pronounced intratumor heterogeneity. Through multiplex spatial analysis, we identified MEKi-induced MHCI upregulation that was specific to GATA6+ tumor cells, which, however, was reduced in proportion due to parallel MEKi-induced mesenchymal cell state switch. By targeted GATA6 degradation, we found a regulatory function for GATA6 in MEKi-induced MHCI expression. Notably, when combined with HDACi, the MEKi-responsive GATA6+ tumor population was preserved, leading to further enhance-ment of MHCI expression and anti-tumor cytotoxicity. This study investigates the cellular interaction between tumor cells undergoing cell state switches with cells of the immune compartments upon therapeutic perturbation, unveiling a GATA6-dependent anti-tumor cytotoxic response. Therapeutic strategies that aim at restoring and augmenting the abundance of GATA6+ tumor cell states may enhance immune recognition of PDAC cells and potentiate anti-tumor cytotoxic effects.

## Results

### GATA6 expression associates with immunoreactive stroma in human PDAC

We assessed GATA6 expression specifically in tumor epithelial cells in a tissue microarray (TMA) from PDAC resections ($n = 143$) by IHC. Epi-thelial regions were manually annotated for each of the 594 cores by a trained pathologist before GATA6 expression was digitally quantified. All cores per patient were averaged, and patients were categorized into GATA6high, GATA6intermediate, or GATA6low groups by expression quan-tiles (Fig. S1a, Supplementary Dataset 1). The abundance of a panel of immune markers that was previously assessed in the cohort[29] was then compared as a function of GATA6 expression in tumor cells. We found significantly higher levels of CD4+, CD20+, FOXP3+, CD68+, and tumor-infiltrating CD3+ and CD8+ cells in the high GATA6 expression group (Fig. 1a, Supplementary Dataset 1). Besides, a positive correlation was observed between tumor GATA6 and E-cadherin expression (Fig. S1a, Supplementary Dataset 1), supporting the specific expression of GATA6 in the classical PDAC subtype.

To dissect the role of GATA6 in tumor cells and their associated stroma, we analyzed the RNA-sequencing dataset of Maurer et al.[30] (GSE93326, $n = 65$), where PDAC malignant epithelium and stroma were procured by Laser Capture Microdissection (LCM). Patients were dichotomized into high ($n = 32$) and low ($n = 32$) tumor GATA6 expres-sion groups (cutoff: median). Gene set enrichment analysis (GSEA) was performed and revealed strong immunological features in the high GATA6 epithelium group and their paired stroma (Fig. 1b, Supplemen-tary Dataset 2). Notably, gene sets including inflammatory response, allograft rejection, TNFα via NFκB, IL2-STAT5, cytokine cytokine recep-tor interaction signaling are all commonly enriched in both GATA6high epithelium and the paired stroma. In addition, IL6-STAT3, leukocyte migration signaling are enriched in GATA6high epithelium, and leukocyte transendothelial migration, antigen processing and presentation, inter-feron gamma response, and immune response to tumor cell signaling are enriched in GATA6high stroma (Fig. 1b, Supplementary Dataset 2).

Next, we investigated the cellular interaction of GATA6+ tumor cells. We performed CellChat analysis with a public PDAC single-cell RNA-sequencing (scRNA-seq) dataset (GSE212966, $n = 6$), to predict cellular interaction between GATA6+/− tumor cells and different cell types and found that GATA6+ tumor cells demonstrated stronger interaction specifically with CD8+ T cells, CD68+ macrophages and CD20+ B cells than GATA6− counterparts (Fig. 1c). GSEA from a com-parison between GATA6+ and GATA6− tumor cells showed that GATA6+ tumor cells were enriched for genes involved in inflammatory response pathways including TNFα via NFκB, cytokine activity, che-mokine receptor binding and response to chemokines (Fig. 1c, Sup-plementary Dataset 3), which was consistent with the GSEA results in GSE93326 (Fig. 1b, Supplementary Dataset 2).

To validate the cellular interaction of GATA6+ tumor cells with immune cells at spatial level, we performed highly multiplex spatial imaging with Phenocycler on 8 therapy-naive human PDAC samples (Figs. 1d and S1b, c). Among these, 2 were resected PDAC primary tissues, while the remaining 6 cases were tissue microarray (TMA) cores of resected PDAC primary tumors (Fig. S1b, c, Supplementary Table 1). Four regions with heterogeneous GATA6 tumor expression were selected for imaging and analysis (Fig. S1b) from resected PDAC tissues, while whole tissue cores were included for analyses for TMA tissues (Fig. S1c). Average values of different ROIs or tissue cores were used for subsequent statistical analyses. 27 markers were included in the panel (Supplementary Dataset 6), and among which, 10 markers were analyzed for this study, focusing on the status of GATA6+ tumor cells and their interaction with immune cells (Fig. 1d). Proximity ana-lysis was performed to measure the number of various immune populations in close proximity (<50 μM radical distance) of GATA6+ PanCK+ cells and GATA6− PanCK+ cells, respectively (Fig. 1d). The analyses revealed a significantly closer cellular interaction of GATA6+ PanCK+ cells with CD8+ cells, compared to GATA6− tumor cells. Similar trends can be observed for CD4+, CD68+, and CD20+ cells (Fig. 1d), although without statistical significance, given low sample numbers. Since antigen presentation-related gene sets were enriched in both tumor and stroma compartment of high GATA6 tumors (Fig. 1b), we examined and compared the expression of MHCI (HLA-A) in GATA6+ PanCK+ and GATA6− PanCK+ cells, and found that GATA6+ tumor cells express higher levels of MHCI than their negative counterpart (Fig. 1e), implying the higher antigenicity of GATA6+ tumor cells. To better elucidate the status of MHCI+ GATA6+ tumor cells, we examined the proportion of proliferative marker PCNA+ cells within the MHCI+ PanCK+ population, as well as in MHCI+ GATA6+ PanCK+ cells. We observed no significant difference, suggesting that MHCI and GATA6 expression levels may not be associated with the proliferation status of tumor cells (Fig. S1d).

GATA6 is associated with the classical PDAC subtype[1–5]. We examined its co-expression with claudin 18 (CLDN18), which was recently linked to the classical and immunogenic PDAC subtype[5,31]. Additionally, keratin 5 (KRT5), a basal-like subtype marker[32], was included for subtype analysis (Fig. S1e). Analysis of eight PDAC samples revealed a high degree of tumor subtype heterogeneity based on these four markers. Notably, GATA6 and CLDN18 did not co-express with KRT5 in any of the cases, supporting their classification as classical subtype markers. As expected, KRT5 expression was restricted to a small proportion of tumor cells, leaving a substantial fraction of the tumor cell population unclassified. Moreover, not all CLDN18+ classical tumor cells expressed GATA6, and vice versa (Fig. S1e), highlighting intrinsic heterogeneity within the classical subtype. Interestingly, GATA6+ CLDN18− PanCK+ cells, but not GATA6− CLDN18+ PanCK+ cells, were in close proximity with CD8+ T cells (Fig. S1f), suggesting a potential immunogenic role of the GATA6+ tumor subpopulation within classical tumor cells. Therefore, we hypothesized that this GATA6+ tumor subpopulation might be a target population for tumor-immune interactions and therapeutic perturbation.

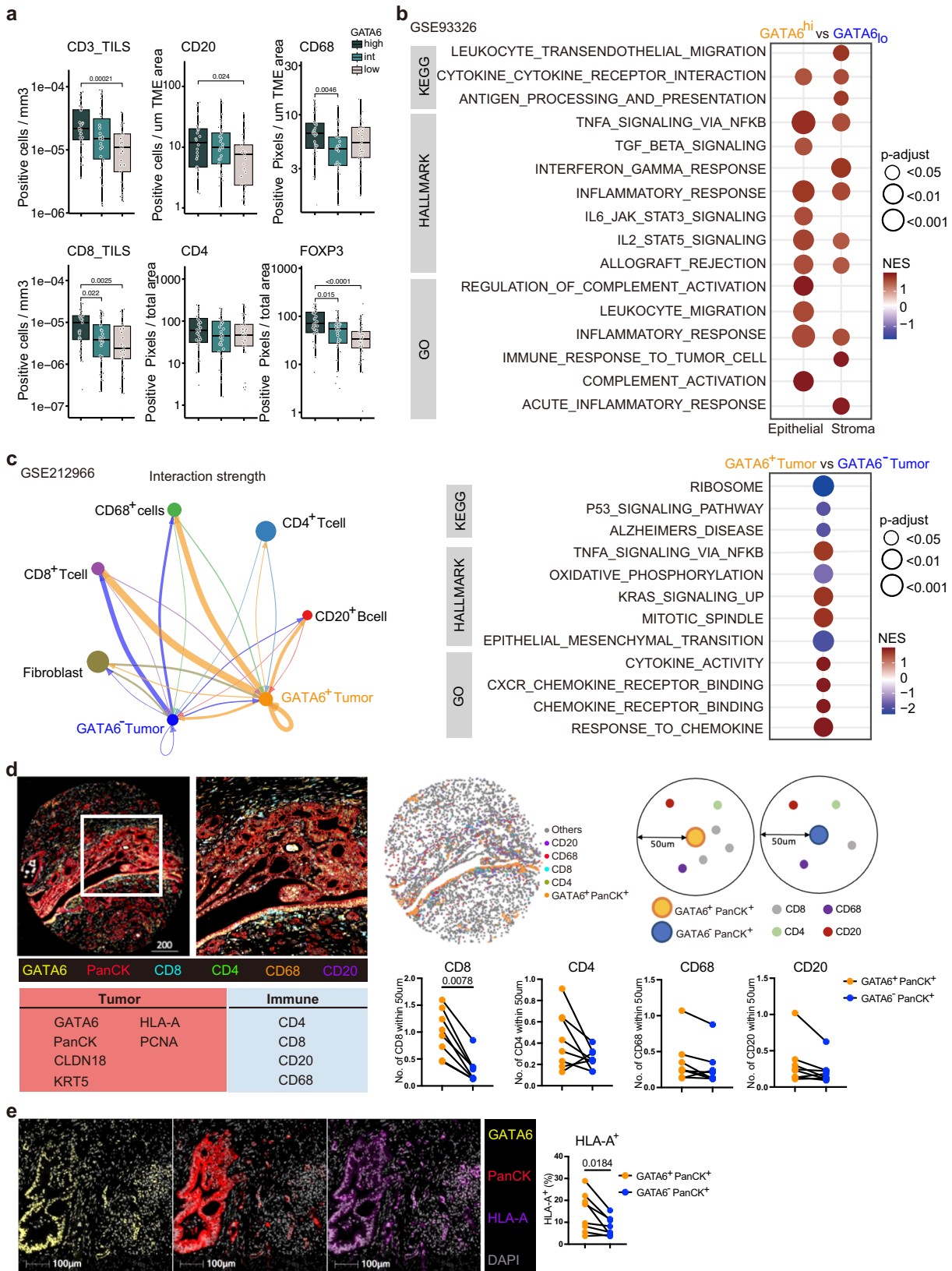

## MEK inhibition induces GATA6⁺ tumor cell-specific MHCI upregulation

We next investigated the impact of targeted MEK inhibition on GATA6 tumor cell expression and the immunological responses of GATA6⁺ tumor cells. Six primary cell lines (511950, 60400, 70301, 60531, 511892, 60590) derived from spontaneous PDAC of six different *Ptf1a^{wt/Cre};Kras^{wt/LSL-G12D};Trp53^{loxP/loxP}* (*CKP*) mice and one primary cell line (110299) from spontaneous PDAC of *Ptf1a^{wt/Cre};Kras^{wt/LSL-G12D};Trp53^{loxPl/R172H}* (*KPC*) mouse[33] were established. In addition, we included two *Kras^{wt/LSL-G12D};Trp53^{wt/LSL-R172H};Pdx1-Cre;Rosa26^{YFP/YFP}*(*KPCY*) derived cell lines 2838c3 and 6694C2, which were tumor cell clones generating tumors that recapitulated T cell-inflamed and non-T cell-inflamed TME

**Fig. 1 | GATA6 expression associates with immunoreactive stroma in human PDAC.** Phenotypic profiles of PDAC patients with differential GATA6 tumor expression at **a** protein and **b** transcriptomic levels. GATA6 expression in tumor epithelial cells in a TMA from PDAC resections (*n* = 143 biological replicates) was assessed by IHC. Epithelial regions were manually annotated for each of the 594 cores by a trained pathologist before GATA6 expression was digitally quantified. All cores per patient were averaged and patients categorized into GATA6^high, GATA6^intermediate, or GATA6_low groups by expression quantiles. CD8, CD4, CD20, CD68, FOXP3 were stained by IHC in resections (*n* = 143 biological replicates) and quantified by QuPath bioimage analysis software. Expression levels of tumor-infiltrating (TILS) CD3 and CD8, CD20, CD68, CD4 and FOXP3 in each group are shown. Box plots show the median (center line), 25th–75th percentiles (bounds of box), and whiskers extending to the minima and maxima values. Individual data points represent independent patients. Statistical significance was calculated by Wilcoxon test. **b** Maurer et al. dataset (GSE93326, *n* = 65 biological replicates). Enrichment was tested against the differential expression profile of epithelium GATA6^hi vs GATA6_lo (*n* = 32 vs *n* = 32 biological replicates, cutoff: median) in epithelium and their paired stroma samples separately. **c** Cellular interaction analysis on GATA6^{+/−} tumor cells in a single-cell RNA-sequencing dataset from Chen et al. (GSE212966, *n* = 6 biological replicates). Left panel: CellChat analysis of cellular interactions among GATA6^{+/−} tumor cells and CD20^+ B cells, CD8^+ and CD4^+ T cells, CD68^+ cells, and Fibroblasts. The line thickness represents the strength of the interaction signals, the direction of the arrows indicates the orientation of the signals. Right panel: GSEA was performed, and selected pathways enriched in GATA6^+ tumor cells when compared to the GATA6^− counterpart were shown. **d** Multiplex imaging by Phenocycler on human PDAC tissues (*n* = 8 biological replicates) demonstrated heterogeneous tumor GATA6 expression associated with infiltration of CD4^+, CD8^+, CD20^+, and CD68^+ cells in the neighborhood (left panel, bar unit: μm). Computational spatial analysis by HALO software showed higher number of immune cells in close proximity (<50 μM radical distance) of GATA6^+ tumor cells than the GATA6^− counterparts (*n* = 8 biological replicates) (right panel). Statistical significance was calculated by two-tailed Wilcoxon matched-pairs signed rank test. **e** GATA6^+ tumor cells (GATA6^+PanCK^+) express higher level of MHCI (HLA-A) than their negative counterpart (GATA6^−PanCK^+). Quantification was done by HALO software (*n* = 8 biological replicates). Bar unit: μm. Statistical significance was calculated by two-tailed Wilcoxon matched-pairs signed rank test.

in vivo, respectively[34]. We measured their GATA6 expression at protein level by flow cytometry (Fig. 2a) and dichotomized the cell lines into GATA6^high (110299, 511950, 2838c3, 60400, 70301) and GATA6_low (60531, 511892, 6694C2, 60590) groups (cutoff: median). We next generated corresponding long-term MEKi-treated counterparts from six of them (three GATA6^high: 511950, 60400, 70301 vs three GATA6_low: 60531, 511892, 60590) upon long-term trametinib treatment as previously described[35] and compared their transcriptomic profiles (Fig. 2b). Notably, we observed enrichment of several immune-related gene sets, including allograft rejection, antigen processing and presentation, IFNα and IFNγ responses upon MEKi treatment only in GATA6^high, but not in GATA6_low cell lines (Fig. 2b, Supplementary Dataset 4).

Impaired antigen presentation machinery and MHCI down-regulation are frequently observed in PDAC and are considered as pivotal mechanisms for tumor immune evasion during the development of drug resistance. We assessed the surface MHCI expression upon MEKi treatment in four GATA6^high and four GATA6_low murine cell lines by flow cytometry. Intriguingly, significant induction of surface MHCI was observed upon MEKi treatment in GATA6^high, but not in GATA6_low cell lines (Fig. 2c). Notably, cell proliferation assays confirmed that there was no significant change in cell proliferation upon treatment (Fig. S2a), suggesting that the increase in MHCI in GATA6^high cell lines was not associated with changes in cell proliferation.

### Targeted GATA6 degradation ablated MEK inhibition-induced MHCI upregulation

To confirm the role of GATA6 in MEKi-induced MHCI expression, we generated two GATA6 knockout cell lines from the GATA6^high cell line 2838c3 (Figs. 2d and S2b). Results show that MEKi-induced MHCI expression was completely abolished in GATA6 knockout cell lines (Fig. 2e), suggesting that MEKi upregulates surface MHCI expression of tumor cells in a GATA6-dependent manner.

Since cells may develop compensatory mechanisms in response to GATA6 loss, to specifically investigate the role of GATA6 in MEKi-induced MHCI expression in PDAC cells, we generated an auxin-inducible degron (AID) knock-in cell line for targeted GATA6 protein degradation (AID-GATA6) and corresponding wild-type (WT) (Figs. 2f and S2c–f). The murine primary PDAC cell line 110299[33] was selected due to its high endogenous GATA6 expression (Fig. 2a). GATA6 degradation was observed after 8 h of incubation with 5-Ph-IAA, with more pronounced degradation after 24 h treatment (Figs. 2g and S2g). Both 110299^WT and 110299^AID-GATA6 cells were treated with 5-Ph-IAA for 24 h, followed by MEKi treatment for an additional 48 h. Upon MEKi treatment, significant MHCI upregulation was detected in 110299^WT cells and 110299^AID-GATA6 cells, which were not treated with 5-Ph-IAA, but not 110299^AID-GATA6 cells treated with 5-Ph-IAA (Fig. 2h), further supporting the role of GATA6 in MEKi-induced MHCI regulation.

### GATA6 is essential for MEKi-induced tumor control and anti-tumor cytotoxicity in vivo

To investigate the role of GATA6 in MEKi-induced effects in vivo, we established syngeneic orthotopic tumor models using one GATA6^high (110299) and one GATA6_low (60590) cell line and treated them with MEKi (Fig. 3a). First, we examined GATA6 expression in the tumors and confirmed that the orthotopic tumors reflected the distinct GATA6 expression levels of the cell lines (Fig. 3b). Upon MEKi treatment, we observed no significant effect on the growth of GATA6_low tumors (60590), while it significantly suppressed the growth of GATA6^high tumors (110299) (Figs. 3c and S3a, b).

Histological analyses of GATA6^high tumors revealed elevated MHCI expression following MEKi treatment (Fig. 3d). Additionally, we observed increased infiltration of CD8^+ and granzyme B (GzmB)^+ cells, with cytotoxicity reflected by higher levels of cleaved caspase-3 (Cl casp3) in MEKi-treated tumors (Fig. 3d). Our previous analysis showed that MEKi-treated GATA6^high tumor cells were enriched for IFNγ response signature (Fig. 2b). Since IFNγ is a key regulator for MHCI expression on tumor cells, we deduced that GATA6 degradation might disrupt IFNγ response that led to ablated MHCI expression on the tumor cells. Therefore, we assessed pSTAT1, a key downstream signaling molecule of IFNγ pathway responsible for regulating MHCI expression[36]. The results showed strong pSTAT1 expression in GATA6^high tumors, with even more intense pSTAT1 signals in MEKi-treated tumors (Fig. 3d). In contrast, GATA6_low tumors, which showed relatively low expression of pSTAT1, exhibited minimal or no expression of MHCI, CD8, or Cl casp3 upon MEKi treatment (Fig. S3c).

To confirm the role of GATA6 in MEKi-induced anti-tumor responses in the 110299 model, 110299^AID-GATA6 cells and 110299^WT control cells were transplanted, and 5-Ph-IAA (auxin) was administered to degrade GATA6 protein only after tumor formation (200–400 mm³) to ensure that GATA6 degradation did not interfere with tumor development or the immune microenvironment before MEKi treatment (Fig. 3e). GATA6 levels were assessed at different time points after 5-Ph-IAA treatment, and GATA6 expression was notably suppressed by more than 50% after 2 days of treatment (Fig. S3d). MEKi treatment was initiated 3 days after GATA6 degradation (Fig. 3e), and strikingly, the MEKi-induced tumor suppression was completely abolished upon GATA6 degradation (Fig. 3f). Tumors were examined histologically, and significant GATA6 degradation in the tumors was confirmed (Fig. 3g). Notably, MEKi-induced increases in MHCI, CD8, Cl casp3, and pSTAT1 were entirely abolished in the GATA6-degraded

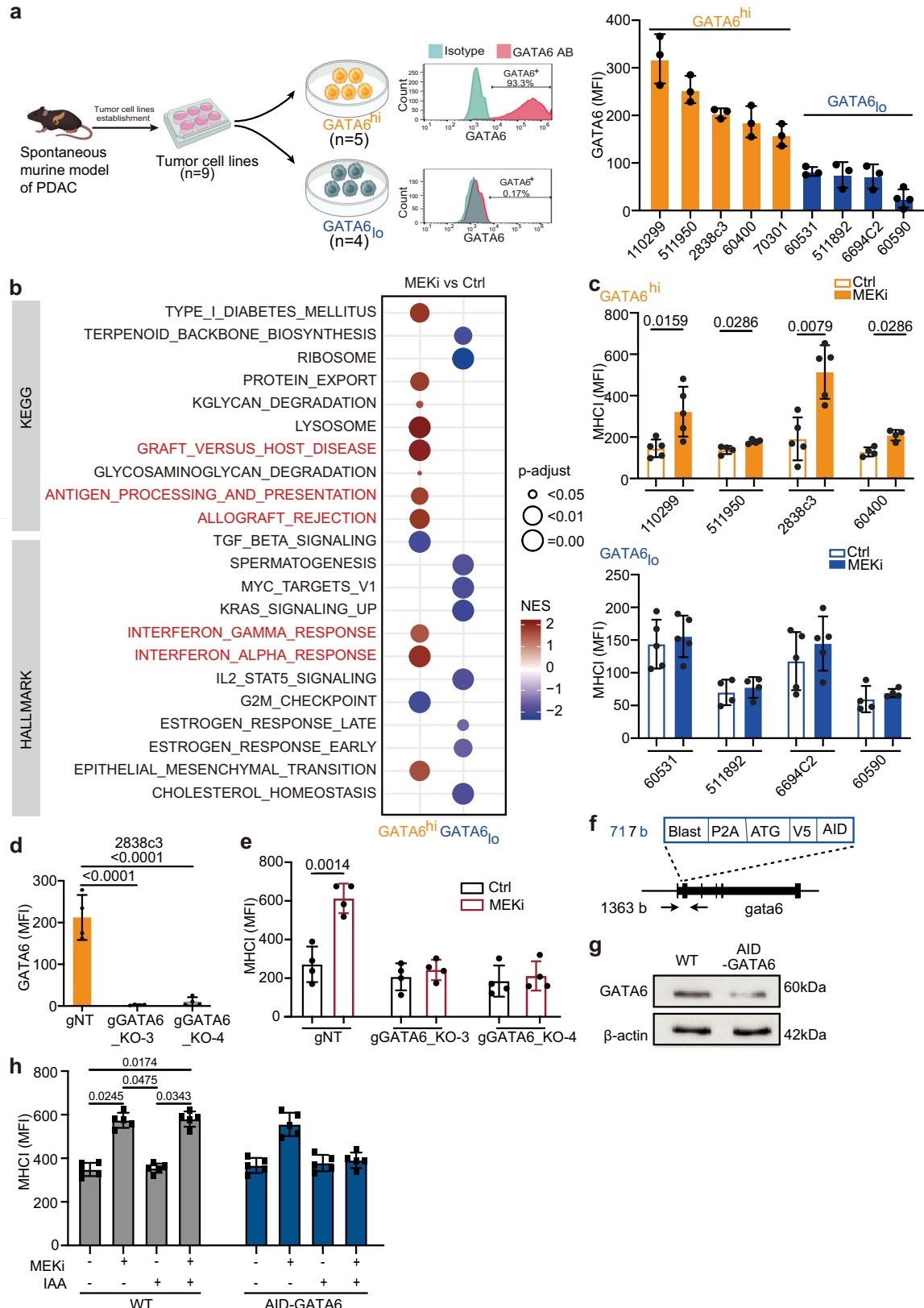

model of 110299$^{AID-GATA6}$ tumors (Fig. 3h), indicating that GATA6 largely contributes to the MEKi-induced anti-tumor responses. It is noteworthy that 110299$^{WT}$ tumors treated with 5-Ph-IAA showed no difference in tumor growth (Fig. S3e).

In addition to CD8$^+$ cells, CD4$^+$ cell infiltration was also enhanced in 110299$^{AID-GATA6}$ tumors following MEKi treatment, although MHCII

levels remained unchanged (Fig. S3f). Strikingly, MEKi-induced CD4$^+$ cell infiltration persisted despite GATA6 degradation (Fig. S3f), indicating that the GATA6-dependent MEKi-induced tumor inhibition is not mediated by CD4$^+$ cells. These findings highlight a specific role for GATA6 in mediating MEKi-induced anti-tumor cytotoxic responses via the IFNγ-MHCI axis.

**Fig. 2 | MEK inhibition induces GATA6-dependent MHCI upregulation. a** Nine primary murine PDAC cell lines were established from spontaneous murine PDAC models (*CKP*, *KPC*, and *KPCY*). Created with BioRender.com. Flow cytometry analysis was used to determine GATA6 protein expression in each line. Representative histograms show GATA6 staining (GATA6 antibody, GATA6 AB, red) versus isotype control (blue) in GATA6 high (GATA6$^{hi}$) and low (GATA6$_{lo}$) expressing cell lines. The cell lines were dichotomized into GATA6$^{high}$ ($n = 5$, orange) and GATA6$_{low}$ ($n = 4$, blue) groups. **b** The parental cells (Ctrl) of three GATA6$^{high}$ cell lines (60400, 511950, 70301) and three GATA6$_{low}$ cell lines (60590, 60531, 511892) were treated with increasing doses of MEKi (trametinib) until 100x of the cells' initial IC50 to generate the corresponding long-term MEKi-treated cells (MEKi). Transcriptomic profiling by RNA sequencing of the parental (Ctrl) and long-term MEKi-treated (MEKi) cells was performed. GSEA of KEGG and HALLMARKs revealed gene sets that were significantly (*P*-adjust < 0.05) different between parental (Ctrl) and long-term MEKi-treated (MEKi) cells within GATA6$^{high}$ and GATA6$_{low}$ groups, respectively. **c** Surface MHCI (H-2Db) expression on control (Ctrl) and MEKi-treated (MEKi) cells of GATA6$^{high}$ (upper panel) and GATA6$_{low}$ group (lower panel) was assessed by flow cytometry ($n = 4$ biological replicates for cell line 60400, 511950, 511892, 60590,

and 5 biological replicates for cell line 110299, 2838c3, 60531, 6694C2). MFI mean fluorescence intensity. Mean ± SD is shown. Statistical significance was calculated by unpaired two-tailed Mann–Whitney test. **d** Flow cytometric analysis of GATA6 expression in GATA6 knockout (KPCY1-CRISPR-GFP v2.1-gRNA-gCas9): gGATA6_KO-3 and gGATA6_KO-4, and negative control (gNT) cell lines ($n = 4$ biological replicates). Statistical significance was calculated by unpaired two-tailed Mann–Whitney test. **e** Surface MHCI (H-2Db) expression on GATA6 knockout cell lines and negative control upon MEKi treatment ($n = 4$ biological replicates). MFI: mean fluorescence intensity. Mean ± SD is shown. Statistical significance was calculated by unpaired two-tailed Mann–Whitney test. **f** Schematic of the knock-in strategy for N-terminal AID-tagged gata6. Shown are components of the knock-in cassette, and the positions of primers for genomic PCR are marked by arrows. **g** Western blot showing GATA6 protein expression in 110299$^{WT}$ and 110299$^{AID-GATA6}$ cells after 24 h treatment with 1 μM 5-Ph-IAA ($n = 1$ biological replicates). **h** Surface MHCI (H-2Db) expression on 110299$^{WT}$ and 110299$^{AID-GATA6}$ cells treated with or without MEKi and/or 1 μM 5-Ph-IAA ($n = 5$ biological replicates). MFI: mean fluorescence intensity. Mean ± SD is shown. Statistical significance was calculated by One-way ANOVA and Kruskal–Wallis test.

## MEKi promotes tumor MHCI expression in GATA6$^+$ PDAC subtype with cell state transition in patient-derived xenografts

While MEKi promotes MHCI upregulation, our previous findings revealed an enrichment of EMT-associated features in long-term MEKi-treated cells derived from *CKP* tumors[35], irrespective of subtype, as indicated by transcriptomic profiling. Given that GATA6 is a known suppressor of EMT in PDAC cells, MEKi may drive tumor cell state transitions by downregulating GATA6 expression. Consistently, analysis of orthotopic tumors in the GATA6$^{high}$ 110299 model also confirmed a reduction in GATA6 expression following MEKi treatment (Fig. S4a).

To confirm the reduction of epithelial features following MEKi treatment in a spontaneous mouse model that closely recapitulates PDAC tumor heterogeneity, *CKP* mice, which develop aggressive endogenous PDAC at 4–6 weeks of age, were treated with MEKi until the study endpoint (Fig. S4b). MEKi-treated tumors exhibited pronounced shrinkage during the first 2 weeks of treatment compared to vehicle controls (Fig. S4c). However, rapid tumor recurrence was observed in all MEKi-treated mice approximately 3 weeks after treatment (Fig. S4c). Notably, E-cadherin expression was significantly reduced in recurrent tumors following MEKi treatment (Fig. S4d), indicating a loss of epithelial lineage features.

To validate these findings in a clinically relevant setting, we analyzed GATA6 and MHCI expression levels in 15 distinct patient-derived xenografts (PDXs) from the CAM-PaC (Integrative Analysis of Gene Functions in Cellular and Animal Models of Pancreatic Cancer) cohort[37] (Fig. 4a). Based on baseline GATA6 expression in tumors without MEKi treatment, the 15 PDXs were stratified into GATA6$^{high}$ ($n = 7$) and GATA6$_{low}$ ($n = 8$) groups (Fig. S5a). MEKi treatment significantly upregulated MHCI expression in GATA6$^{high}$ tumors but not in GATA6$_{low}$ tumors (Fig. 4b), consistent with our previous in vitro findings. Notably, MEKi-treated GATA6$^{high}$ tumors exhibited a reduction in GATA6 expression (Fig. 4c) alongside an increase in the mesenchymal marker KRT81 (Fig. S5b), suggesting a tumor cell state transition. In contrast, no significant changes were observed in GATA6$_{low}$ tumors (Fig. S5c).

Collectively, these findings suggest that while MEKi enhances MHCI expression in GATA6$^{high}$ tumor cells, it concurrently promotes a cell state transition by suppressing GATA6 expression not only in murine but also in human PDAC cells.

## HDAC inhibitors restored GATA6 expression and promoted MHCI expression

Given the critical role of GATA6 in MEKi-induced anti-tumor immune responses, mitigating MEKi-induced EMT while preserving an immune-amendable GATA$^+$ tumor cell population is of therapeutic interest. Histone deacetylase inhibitors (HDACi) have been reported to inhibit

EMT and enhance anti-tumor cytotoxicity by promoting antigen presentation[25,28,38]. Thus, we selected a PDX with high GATA6 expression to assess the combination of the MEKi trametinib with the class I HDACi mocetinostat. While MEKi treatment slightly reduced GATA6 levels, the combination with mocetinostat significantly restored GATA6 expression in tumors (Fig. S5d, upper panel). Notably, while MEKi alone increased MHCI expression, this effect was further enhanced upon combination treatment (Fig. S5d, lower panel).

Given the differential therapeutic and toxicity profiles of various HDACi, we next evaluated a panel of class I (mocetinostat, quisinostat, domatinostat) and class II (tasquinimod, LMK235, ricolinostat) HDAC inhibitors, either alone or in combination with MEKi, for their effects on GATA6 expression in the GATA6$^{high}$ murine PDAC cell line 110299. MEKi treatment significantly reduced GATA6 expression, while the addition of domatinostat largely restored its levels (Fig. 4d). This aligns with previous findings that domatinostat promotes epithelial gene expression in PDAC cells through a BRD4- and MYC-dependent mechanism[25]. These findings were further confirmed by western blot analysis across eight murine PDAC cell lines, where GATA6 restoration was observed in all GATA6$^{high}$ cell lines upon combination treatment with MEKi and domatinostat, but not obvious in GATA6$_{low}$ cell lines (Fig. 4e). We also examined the epithelial marker E-cadherin and the mesenchymal marker N-cadherin. While MEKi consistently induced EMT across all cell lines, its combination with domatinostat effectively reversed this effect in GATA6$^{high}$ cell lines by restoring E-cadherin and suppressing N-cadherin expression (Fig. S5e). These findings suggest that HDAC inhibition restores GATA6 expression by counteracting MEKi-induced cell state transitions.

In addition to EMT inhibition, previous studies have shown that HDACi promotes anti-tumor cytotoxicity by enhancing antigen presentation. To further investigate this, we assessed the effects of class I (mocetinostat, quisinostat, domatinostat) and class II (tasquinimod, LMK235, ricolinostat) HDACi, alone or in combination with MEKi, on MHCI expression in GATA6$^{high}$ and GATA6$_{low}$ murine PDAC cell lines. All HDACi treatments increased surface MHCI expression, with class I HDACi generally inducing a greater effect than class II HDACi (Figs. 4f and S5f, g). Notably, while MEKi-induced MHCI upregulation was restricted to GATA6$^{high}$ cell lines, HDACi treatment increased MHCI expression also in GATA6$_{low}$ cell lines (Figs. 4f and S5f, g). Furthermore, combination treatment with MEKi further augmented surface MHCI levels across most of the cell lines, with the most pronounced increase observed for the MEKi and domatinostat combination (Figs. 4f and S5f, g).

These findings suggest that HDACi not only counteract MEKi-induced GATA6 suppression, but also enhance MHCI expression on tumor cells, thereby potentially augmenting anti-tumor immunity.

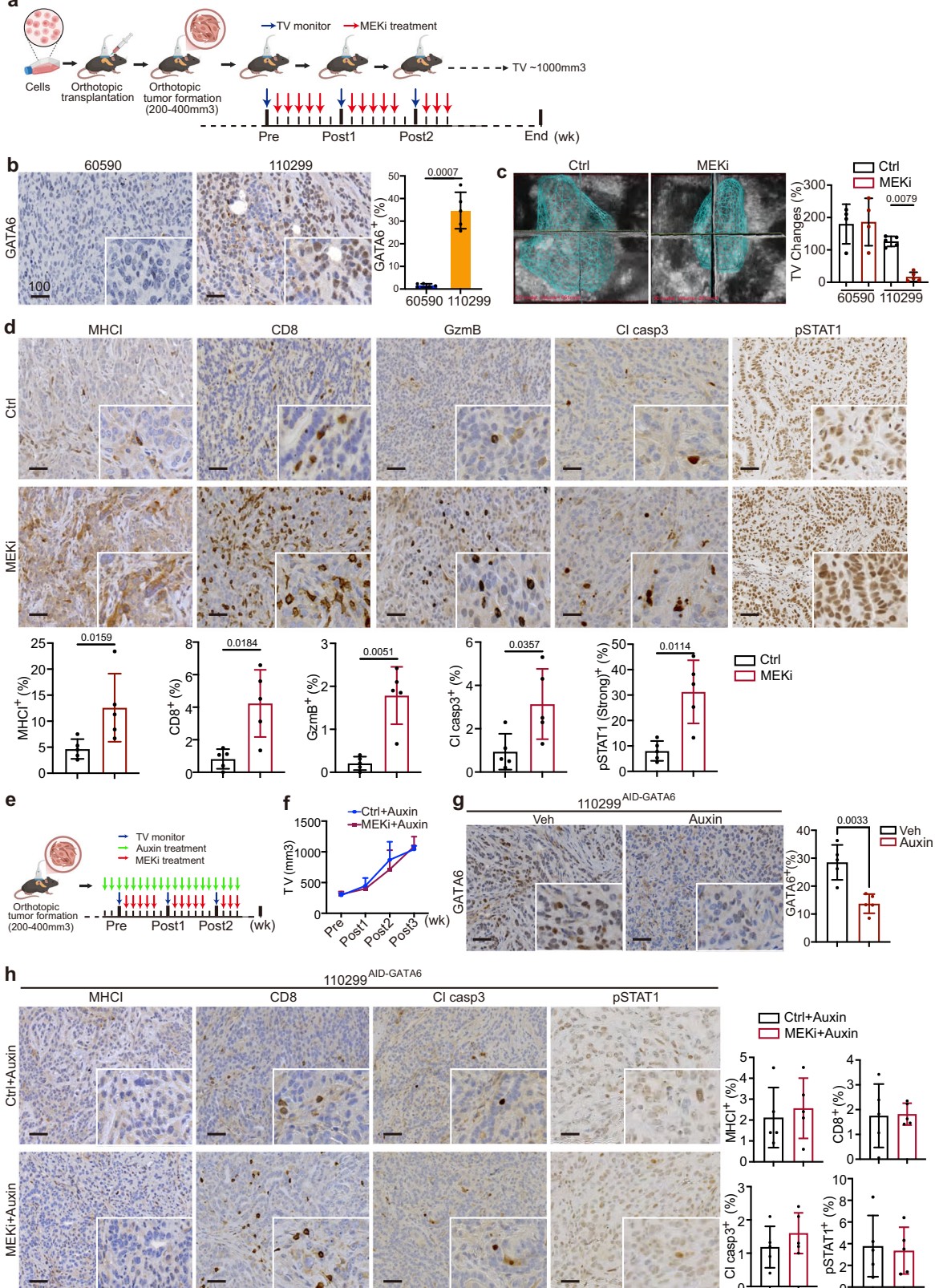

**Combination of MEKi and HDACi promotes tumor-specific T cell anti-tumor cytotoxicity specifically in GATA6$^{high}$ tumor cells in vitro**

Next, we investigated whether the MEKi- and HDACi-induced upregulation of MHCI expression translates into enhanced anti-tumor cytotoxicity. Given the typically low mutational burden of PDAC, which limits the availability of neoantigens, we utilized a next-generation dual recombination (*Cre/Lox*; *Flp/FRT*) mouse model, *FKPC2GP*, which enables temporally and spatially controlled expression of the model tumor antigen LCMV-gp33 (Fig. 5a)[39-41].

**Fig. 3 | GATA6 is essential for MEKi-induced tumor control and anti-tumor cytotoxicity in vivo. a** Schematic diagram showing the experimental workflow. Orthotopic tumors derived from GATA6$_{low}$ (60590) or GATA6$^{high}$ (110299) cell lines were monitored weekly by 3D ultrasound imaging. MEKi treatment was initiated when tumor volume (TV) reached 200–400 mm³, and experiments were terminated when tumor volume (TV) reached approximately 1000 mm³. Created with BioRender.com. **b** IHC staining of GATA6 in 60590 and 110299 tumors without treatment. The percentage of GATA6$^+$ cells out of total cells was quantified by HALO software (n = 5 independent mice per group). Data are presented as mean values ± SD. Individual data points represent independent mice. Statistical significance was calculated by unpaired two-tailed Mann–Whitney test. **c** Tumor volume (TV) dynamics following MEKi treatment. Left panel: representative 3D ultrasound images of 110299 tumors at post 1 week with or without MEKi (Ctrl: vehicle control) treatment. Right panel: Changes in tumor volume from pre- to post 1 week treatment with or without MEKi in both models (n = 4 independent mice for the 60590 model, and n = 5 independent mice for the 110299 model). Data are presented as mean values ± SD. Individual data points represent independent mice. Statistical significance was calculated by One-way ANOVA and Kruskal–Wallis test. **d** IHC staining of MHCI, CD8, GzmB, cleaved caspase-3 (Cl casp3), and phosphorylated STAT1 (pSTAT1) in 110299 tumors with or without MEKi treatment. The proportion of marker-positive cells out of total cells quantified by HALO software is

shown below (n = 4 independent mice for the 60590 model, and n = 5 independent mice for the 110299 model). Data are presented as mean values ± SD. Individual data points represent independent mice. Statistical significance was calculated by unpaired two-tailed Mann–Whitney test. **e** Schematic diagram of the in vivo MEKi treatment in targeted GATA6 degradation model. Created with BioRender.com. Orthotopic tumors were established using 110299$^{AID\text{-}GATA6}$ cells or 110299$^{WT}$ cells. GATA6 degradation was induced by daily administration of 5-Ph-IAA (auxin) after tumor formation. MEKi treatment was started 3 days after auxin administration. **f** Tumor growth curves showing the ablation of MEKi-induced tumor inhibition of 110299 orthotopic tumors upon GATA6 degradation (n = 5 independent mice for each group). **g** IHC staining confirmed GATA6 degradation after auxin treatment in 110299$^{AID\text{-}GATA6}$ tumors. Right panel: Quantification of GATA6$^+$ cells out of total cells by HALO software (n = 5 independent mice for each group). Data are presented as mean values ± SD. Individual data points represent independent mice. Statistical significance was calculated by unpaired two-tailed Mann–Whitney test. **h** IHC analysis in GATA6-depleted tumors showing abolished MEKi-induced increases in MHCI, CD8, Cl casp3, and pSTAT1. Right panel: Quantification of marker-positive cells out of total cells by HALO software (n = 5 independent mice for each group). Data are presented as mean values ± SD. Individual data points represent independent mice. Statistical significance was calculated by unpaired two-tailed Mann–Whitney test. Scale bar: μm.

To assess tumor-specific cytotoxicity, we established four low-passage *FKPC2GP*-derived primary cell lines: GP58, GP82, and GP99, which express LCMV-gp33 upon tamoxifen induction in vitro, and GP2838c3, which constitutively expresses LCMV-gp33. Following treatment with MEKi and HDACi, LCMV-gp33$^+$ tumor cells were co-cultured with LCMV-gp33-reactive T cells isolated from the spleens of P14-TCR-Tg mice (Fig. 5b).

Baseline GATA6 expression levels were evaluated across these cell lines, revealing that GP82 and GP2838c3 exhibited higher GATA6 expression compared to GP99 and GP58 (Fig. 5c). Based on these findings, we classified GP82 and GP2838c3 as GATA6$^{high}$ models and GP99 and GP58 as GATA6$_{low}$ models for subsequent analyses.

Consistent with our earlier observations, GATA6$^{high}$ GP82 and GP2838c3 cells exhibited a significant increase in surface MHCI expression following combined MEKi and HDACi treatment (Fig. 5d). MHCI expression was also upregulated by HDACi and combined MEKi and HDACi treatment on GATA6$_{low}$ GP99 and GP58 cells, although the magnitude of increase was smaller (Fig. 5d).

To evaluate the functional impact of these changes on anti-tumor immunity, all four LCMV-gp33-expressing tumor cells were co-cultured with LCMV-gp33-reactive T cells isolated from the spleens of P14-TCR-Tg mice. Combined MEKi and HDACi treatment led to a marked accumulation of LCMV-gp33-specific T cells, accompanied by prominent tumor cell death in GATA6$^{high}$ co-cultures (Fig. 5e). Propidium iodide (PI) staining and flow cytometric analysis confirmed that cytotoxicity was significantly elevated in LCMV-gp33-induced GATA6$^{high}$ cells upon treatment (Fig. 5e).

In addition, we assessed the activation status of LCMV-gp33-reactive T cells within the co-culture system. Notably, GzmB and PD1 expression was significantly increased in T cells co-cultured with GATA6$^{high}$ tumor cells treated with MEKi and HDACi, further supporting enhanced T cell activation and tumor-specific cytotoxicity (Fig. 5e). Notably, similar trends could also be observed in GATA6$_{low}$ co-cultures, although the cytotoxic effects were less prominent than in the GATA6$^{high}$ co-cultures (Fig. 5f). Together, these findings demonstrate that combined MEKi and HDACi treatment induces MHCI upregulation in PDAC cells, which translates into enhanced tumor-specific T cell cytotoxicity.

**MEKi and HDACi combination treatment enhances CD8$^+$ T cell-mediated tumor suppression in orthotopic mouse model**
To validate the anti-tumor cytotoxic effects of combined MEKi and HDACi treatment, we administered the combination therapy in an

orthotopic mouse model derived from the GATA6$^{high}$ cell line 110299 (Fig. S6a). Besides, in order to investigate the involvement of CD8$^+$ T cells in the anti-tumor response elicited by the combination therapy, we pre-treated mice with a CD8-depleting antibody prior to administering the combination regimen (Fig. S6b).

Upon combined MEKi and HDACi treatment, tumor growth was markedly suppressed (Fig. S6c). Despite a slight reduction of GATA6 when compared to control tumors, the high proportion of GATA6$^+$ tumor cells remained responsive to the MEKi and HDACi combination therapy, exhibiting significant upregulation of MHCI, increased CD8$^+$ T cell infiltration, and elevated cl casp3 expression (Fig. S6d). These findings suggest that the MEKi and HDACi combined treatment preserves the GATA6-expressing tumor population, which responds by enhancing MHCI expression, ultimately leading to increased CD8$^+$ T cell-mediated cytotoxicity.

Notably, the tumor-suppressive effect of the MEKi and HDACi combined treatment was fully abrogated in CD8-depleted mice (Fig. S6c). Histological analysis confirmed effective CD8$^+$ T cell depletion following anti-CD8 antibody treatment (Fig. S6d). Importantly, tumor cell GATA6 expression remained largely unchanged, while MHCI expression stayed elevated following MEKi and HDACi combination therapy, regardless of CD8 status, suggesting that MHCI upregulation is independent of CD8$^+$ T cell activity (Fig. S6d). In contrast, increase of Cl casp3 after MEKi and HDACi combination therapy was abolished when CD8$^+$ T cells were depleted, further supporting the critical role of CD8$^+$ T cells in mediating the anti-tumor effects of the combined MEK and HDAC inhibition.

**Combination of MEKi and HDACi prolongs survival of *CKP* mice with restored GATA6 and MHCI expression**
We next assessed the efficacy of a combined MEK and HDAC inhibition strategy in a highly aggressive, endogenous PDAC mouse model to reflect better the immunosuppressive, desmoplastic, and heterogenic features of human disease and evaluate its potential clinical relevance. *CKP* mice, aged 6 weeks and presenting measurable advanced tumors, were treated with or without MEKi and/or HDACi until the experimental endpoint (Fig. 6a). While both monotherapies significantly extended survival compared to vehicle-treated controls, the combination treatment further enhanced survival relative to either monotherapy (Fig. 6b).

Tumor volumes assessed by 3D ultrasound imaging showed only 4 out of 6 mice treated with MEKi and none of HDACi-treated mice exhibited tumor shrinkage (Fig. 6c). In contrast, all mice

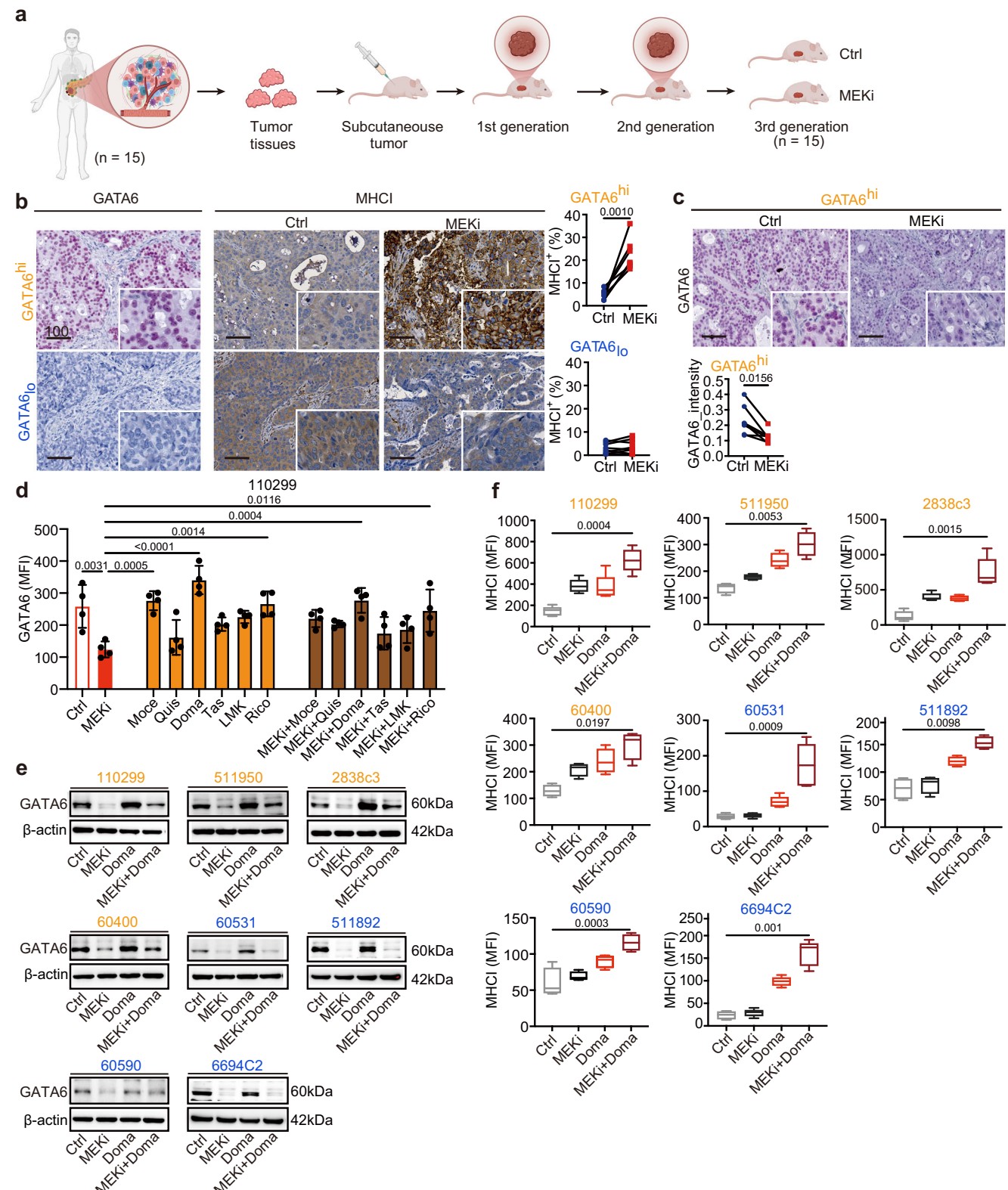

treated with the combination regimen (*n* = 7) showed a sustained reduction in tumor volume throughout the treatment period (Fig. 6c). Importantly, tumor control was maintained in all combination-treated mice at the experimental endpoint, and these animals were euthanized due to weight loss rather than tumor progression (Fig. S7a). Tumor volume changes could not be evaluated in 3 control mice due to their short survival time (less than 1 week after study entry) (Fig. 6c).

To investigate the mechanisms underlying the survival benefit of combination therapy, we compared the immune-related transcriptomic profiles of *CKP* tumors treated with or without MEKi and/or HDACi using the NanoString PanCancer Immune Profiling Panel. Gene set variation analysis (GSVA) revealed that MEKi alone induced the expression of several immune-related gene sets, including antigen receptor-mediated signaling, T cell receptor signaling, type II interferon activation, T cell activation, and cytotoxicity (Fig. 6d), suggesting

**Fig. 4 | MEKi reduces GATA6 expression that is restored by HDAC inhibitors with augmented MHCI expression. a** Patient-derived xenografts (PDXs) from 15 PDAC patients were transplanted subcutaneously into athymic nude mice, and the mice of third generation were used for MEKi treatment. When tumor volume reached 200 mm$^3$, the mice were treated with vehicle control (Ctrl) or MEKi. Tumors were harvested after 4 weeks of treatment. Created with BioRender.com. **b** IHC staining of GATA6 and MHCI of the xenograft tumors was performed, and the proportion of marker-positive cells out of total cells was quantified by Definiens software. Xenografts were dichotomized into GATA6$^{high}$ (GATA6$^{hi}$, $n = 7$ independent mice) and GATA$_{low}$ (GATA$_{lo}$, $n = 8$ independent mice) expression groups using median GATA6$^+$ cells (% total) as cutoff. MHCI$^+$ cells (% total) were quantified by Definiens and compared between vehicle control and corresponding MEKi-treated mice in GATA6$^{high}$ and GATA$_{low}$ expression groups, respectively. Statistical significance was calculated by two-tailed Wilcoxon matched-pairs signed rank test. **c** IHC staining showing reduction of GATA6 expression in MEKi-treated tumors in GATA6$^{high}$ expression group. Signal intensities were quantified by Definiens and compared between vehicle control and MEKi-treated mice groups ($n = 7$ independent mice for each group). Statistical significance was calculated by two-tailed Wilcoxon matched-pairs signed rank test. **d** Flow cytometry analysis of GATA6 expression in 110299 cells treated with MEKi, class I HDAC inhibitors (mocetinostat, quisinostat, domatinostat), class II HDAC inhibitors (tasquinimod, LMK235, ricolinostat), or their combinations ($n = 4$ independent experiments for each cell line). Data are presented as mean values ± SD. Individual dots represent independent biological replicates. Statistical significance was calculated by One-way ANOVA, Kruskal–Wallis test. **e** Western blot analysis of GATA6 protein levels in four GATA6$^{high}$ (orange: 110299, 511950, 2838c3, 60400) and four GATA6$_{low}$ (blue: 60531, 511892, 60590, 6694c2) murine PDAC cell lines following treatment with MEKi and/or domatinostat; β-actin served as a loading control ($n = 1$ independent experiment). **f** Surface MHCI (H-2Db) expression in the four GATA6$^{high}$ (orange) and four GATA6$_{low}$ (blue) murine PDAC cell lines treated with MEKi and/or domatinostat, assessed by flow cytometry ($n = 4$ independent experiments for cell line 511950, 60400, 511892, 60590, and 5 independent experiments for cell line 110299, 2838c3, 60531, 6694C2). Data are presented as box plots: the center line indicates the median, the bounds of the box indicate the 25th and 75th percentiles, and whiskers extend to minima and maxima. One-way ANOVA and Kruskal–Wallis test were used. Ctrl: vehicle control; MEKi: trametinib; Moce: mocetinostat; Quis: quisinostat; Doma: domatinostat; Tas: tasquinimod; LMK: LMK235; Rico: ricolinostat. MFI mean fluorescence intensity. Scale bar: μm.

its role in promoting T cell-mediated immunity. HDACi alone had a more subtle effect on the tumor immune microenvironment, inducing pathways only related to antigen processing and type II interferon regulation (Fig. 6d). Notably, combined MEKi and HDACi treatment activated a broader range of immune pathways, including those enriched in both monotherapies (Fig. 6d).

To further elucidate the differential effects of MEKi alone versus combination treatment, we compared the immune-related transcriptomic profiles and found that the combination therapy significantly upregulated genes associated with T cell function and antigen presentation (Fig. S7b,c). Additionally, the combination group exhibited higher scores in pathways related to antigenicity, such as MHC and antigen processing (Fig. S7b, c), suggesting that combined MEKi and HDACi enhances tumor antigenicity compared to MEKi alone.

We next assessed GATA6 expression in *CKP* tumors. Notably, GATA6 expression, which was suppressed in MEKi-treated tumors, was restored following combination treatment with domatinostat (Fig. 6e), suggesting that HDACi mitigated the MEKi-induced loss of classical-type tumor cells.

We then performed multiplex immunofluorescence (mIF) staining for GATA6 and MHCI in *CKP* tumors. Strong co-expression of these two molecules was observed, particularly in the combination treatment group (Fig. 6f). Quantitative analysis revealed that, upon MEKi treatment alone, the majority of GATA6$^+$ tumor cells were also MHCI$^+$, though GATA6 expression was significantly reduced. Importantly, combination treatment with HDACi significantly restored GATA6$^+$ tumors, with the majority remaining MHCI$^+$ (Fig. 6f, right panel). These findings suggest that HDACi suppresses the MEKi-induced transition and preserves the GATA6$^+$ classical cell state of tumor cells, which are responsive to MEKi with proficient MHCI upregulation. This mechanism enhances the overall antigenicity of the tumors, thereby promoting anti-tumor immune responses.

### Enhanced tumor MHCI expression induced by combined MEK and HDAC inhibition is associated with increased cytotoxic T-cell infiltration and tumor apoptosis in vivo

We next examined the effect of treatment on T cell-mediated anti-tumor activity. IHC analysis revealed a significant increase in infiltrating CD8$^+$ T cells in tumors treated with the combination regimen compared to the other groups (Fig. 7a). Notably, the T cell activation marker PD1 and the cytotoxic marker GzmB were both significantly elevated in combination-treated tumors (Fig. 7a). The enhanced anti-tumor cytotoxicity observed with combination treatment was further supported by increased Cl casp3 levels, with no significant changes seen in the single treatment (Fig. 7b).

We validated the increased T cell-mediated cytotoxicity in combination-treated tumors by performing mIF for MHCI, GzmB, Cl casp3, and PanCK as a tumor cell marker, along with computational spatial analysis. In tumor regions with high MHCI expression, we observed abundant GzmB$^+$ cells and increased Cl casp3 expression. In contrast, regions with low MHCI expression showed reduced levels of these markers (Fig. 7c). Co-expression analysis revealed that MHCI$^+$ tumor cells significantly expressed more Cl casp3 compared to MHCI$^-$ tumor cells (Fig. 7d). Additionally, spatial analysis demonstrated that a higher number of GzmB$^+$ cells were in close proximity (<150 μm radical distance) to MHCI$^+$Cl casp3$^+$ cells compared to MHCI$^-$ Cl casp3$^+$ cells (Fig. 7e), supporting the critical role of MHCI in tumor immunogenicity and anti-tumor cytotoxicity in PDAC.

Our results demonstrate that MEKi induces GATA6-dependent MHCI upregulation in tumor cells. However, this effect is diminished by a treatment-induced transition to a different cell state, resulting in the loss of GATA6 expression. Combining MEKi with HDACi preserves the GATA6$^+$ tumor cell population, leading to further enhancement of MHCI expression and increased anti-tumor cytotoxicity (Fig. 8).

## Discussion

PDAC exhibits dynamic cellular heterogeneity and treatment-induced adaptations in both tumor and non-malignant cells[29,42], complicating treatment strategies and therapeutic resistance patterns[3,4,43,44]. While GATA6 is a marker of classical tumor subtypes[1,43], it also plays a crucial role in regulating EMT, tumor dissemination, and interactions with the immune microenvironment[1], potentially influencing therapeutic response.

A recent study demonstrated in an elegant dual-recombinase mouse model for GATA6 deletion at late states of KRas$^{G12D}$-driven pancreatic tumorigenesis (GATA6$^{LateKO}$), that the loss of GATA6 was associated with basal and metastatic phenotypes, as well as features of immune escape in mouse and human PDAC cells[12]. Congruent with our findings, the group observed a significant downregulation of MHCI genes in GATA6$^{LateKO}$ murine cells[12]. These findings prompted us to speculate that the restoration of the GATA6$^+$ tumor cell state may be a valid strategy to increase tumor immunogenicity and revert EMT-associated cell features. However, how tumor cells with distinct GATA6 expression and dependent cell phenotypes operate and interact within the local TME, including perturbation-induced dynamics, is not clearly understood.

Here, we demonstrate in vitro that MEKi increases MHCI expression in PDAC cell lines of classical subtype. Notably, in endogenous highly aggressive and desmoplastic tumors of genetically engineered mice, overall MHCI expression does not significantly change upon

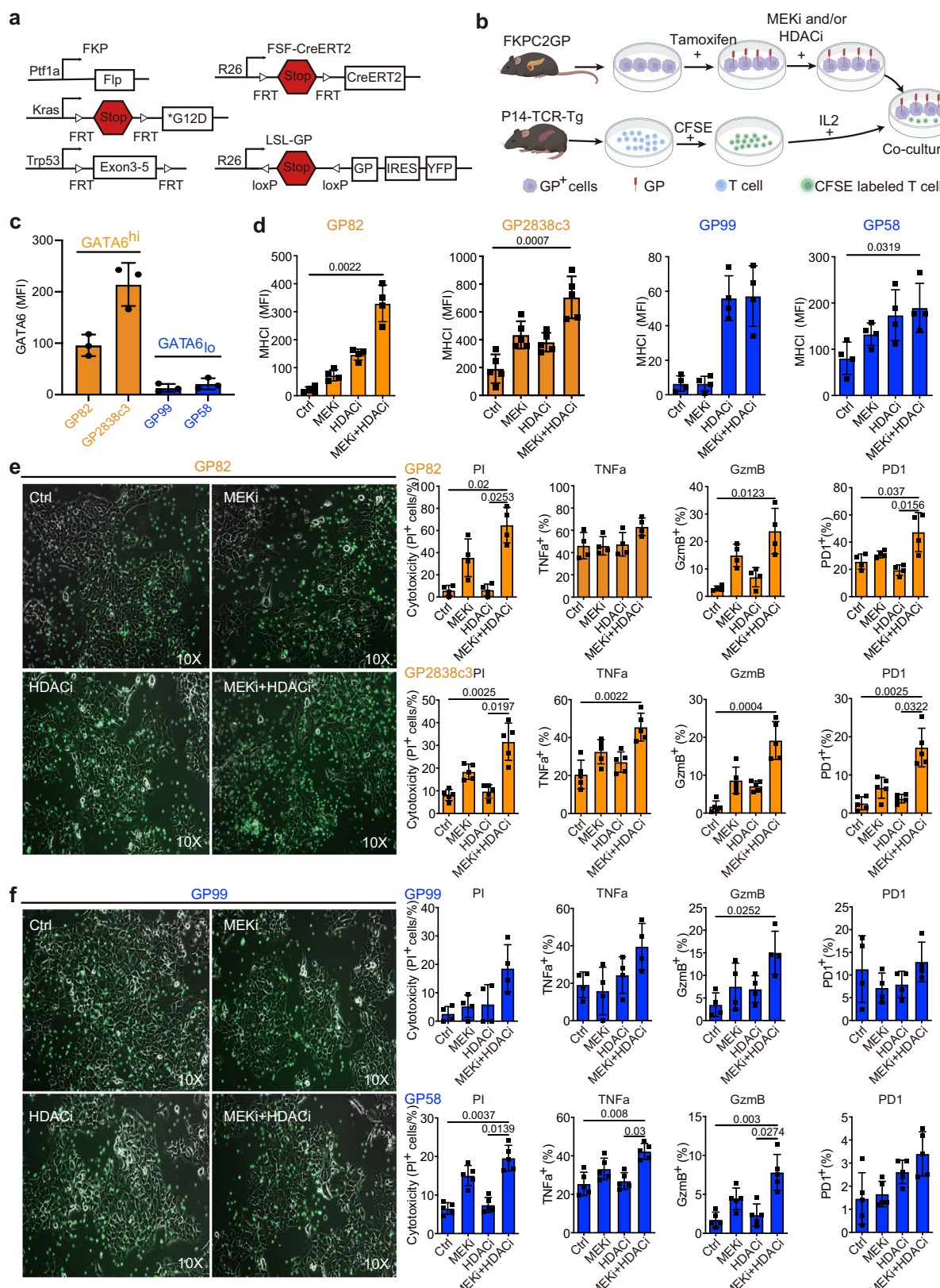

MEKi treatment. The *CKP* model recapitulates human PDAC in terms of complex spatial tumor-stromal tissue heterogeneity, where the co-existence of different tumor cell phenotypes with numerous fibroblast and immune cell subsets in the same tumor is frequently observed[29]. Through multiplex imaging analysis, we confirm the GATA6+ tumor-specific MHCI upregulation upon MEKi at cellular level in the

heterogeneous *CKP* tumors. However, MEKi reduces the proportion of GATA6+ classical tumor cells, which compromises the overall beneficial effect on MHCI expression. Importantly, upon combined treatment with class I HDACi domatinostat, GATA6+ classical tumor cells with augmented MHCI expression are significantly restored. Multiplex spatial analysis also revealed the increased apoptosis of MHCI+ tumor

**Fig. 5 | MEKi treatment promotes tumor-specific T cell anti-tumor cytotoxicity in vitro. a** Genetic strategy to induce spatially and temporally controlled GP33 expression by tamoxifen-mediated activation of *CreERT2* in pancreatic cells harboring mutant *KrasG12D* and loss of *Tp53*. *Ptf1a^{wt/flp}*;*Kras^{wt/FSF-G12D}*;*TrpS3^{tm1.1Dgk}* (*FKP*) mice crossed to *Gt(ROSA)26^{Sortm3(CAG-Cre/ERT2)Dsa}*(*R26FSF-CAG−CreERT2*) and *Gt(ROSA) 26^{SortmloxP-STOP-loxP-GP-IRES-YFP}*(*R26LSL-GP*) strains to generate *FKPC2GP* mice. **b** Experimental setup for co-culture of treated GP, LCMV-gp33-expressing cell lines derived from *FKPC2GP* tumor, and the gp33 reactive T cells isolated from the spleen of *P14-TCR-Tg* mice. The treatment follows the experiments of *CKP* cell lines. Created with BioRender.com. **c** Flow cytometric analysis of GATA6 expression in four GP cell lines (GP82, GP2838c3, GP99, GP58, *n* = 3 independent experiments). GP82 and GP2838c3 are defined as GATA6^{high}, and GP99 and GP59 are defined as GATA6_{low}. **d** Surface MHCI (H-2Db) expression on GP mouse cell lines 72 h treatment as assessed by flow cytometry (*n* = 4 independent experiments for cell line GP82, GP99, GP58, and 5 independent experiments for cell line GP2838c3). MFI:

mean fluorescence intensity. Mean ± SD is shown. One-way ANOVA, Kruskal–Wallis test were used. Treated GATA6^{high} (**e**) and GATA6_{low} (**f**) GP cells were co-cultured with LCMV-gp33-reactive T cells (CFSE-labeled, green) for 2 days. Left panel: Microscopic images of treated GP cells and LCMV-gp33-reactive T cells (CFSE-labeled, green) after 2 days of co-culture. Right panel: Cytotoxicity level (PI^+ %) of GP cells upon co-culture with LCMV-gp33-reactive T cells assessed by flow cytometry. Percentage of CD8^+ cells that are positive for cytotoxic markers TNFa, GzmB, and activation PD1 (% in total T cells) assessed by flow cytometry (*n* = 4 independent experiments performed with LCMV-gp33-reactive T cells isolated from 4 different mice for cell line GP82, GP99, and *n* = 5 independent experiments performed with LCMV-gp33-reactive T cells isolated from 5 different mice for cell line GP2838c3 and GP58). MFI: mean fluorescence intensity. Mean ± SD is shown. One-way ANOVA, Kruskal–Wallis test were used. Ctrl: vehicle control; MEKi: trametinib; HDACi: domatinostat.

cells that closely interact with more cytotoxic T cells (Fig. 7d), providing a potential mechanism underlying the improved survival of combination treatment. Indeed, the spatial data allow unprecedented insight into how cellular dynamics and heterogeneity affect tumor responses to treatment, and can aid in the rational and mechanistic-driven design of combinatory treatment strategies for clinical translation[45,46].

HDAC is able to induce anti-tumor responses not only in cell autonomous manner by suppressing EMT-related gene programs[25,26,47,48], but also non-cell autonomously by targeting non-malignant corrupted cells to achieve an anti-tumor immune responses[27,49,50]. Therefore, HDAC inhibition has been considered as a promising strategy to target adaptive resistance in PDAC. Currently, the HDAC inhibitors panobinostat (pan-class I, II, IV) and romidepsin (class I) are FDA-approved in non-PDAC entities, and several others are under investigation. However, successful clinical development of HDAC inhibitors has been limited by low on-target specificity and treatment-associated toxicities, e.g., cardiotoxicity and myelosuppression[51,52]. Indeed, toxicity was observed in our study with the combined MEKi and HDACi treatment in *CKP* mice, leading to treatment termination due to significant weight loss, despite small tumor volumes (Fig. S7a). Recently, an HDAC6 inhibitor, KA2507, with selective target engagement, better safety profile, as well as enhancement in anti-tumor immune response, was reported in a phase I clinical trial in patients with advanced solid tumors[53]. The combination of HDAC6 inhibitors with other immuno-modulatory drugs may, therefore, represent a potential avenue for future clinical exploration of such an epigenetic-based strategy. We observed an increase in tumor MHCI expression induced by the HDAC6 inhibitor ricolinostat alone, and/or in combination with MEKi in cultured tumor cells. Although the magnitude of increase was not as strong as those exerted by class I HDAC inhibition with domatinostat, the functional effect of HDAC6 inhibitors can be potentially increased with improved pharmacokinetic and pharmacodynamic properties. Further efforts are needed to develop therapeutic strategies combining optimized HDAC inhibitors with RAS/MAPK targeting approaches and immunotherapy for future clinical exploration.

Despite intensive efforts, immune checkpoint inhibitors (ICIs), either alone or in combination with conventional therapies, have yielded limited efficacy in PDAC. This failure underscores the complexity and heterogeneity of PDAC and so far unresolved challenges, including T cell exhaustion and exclusion, dysfunctional antigen presentation machinery, stromal barriers, abundance of immunosuppressive cells, and secreted factors. Our results demonstrate that concurrent MEK and HDAC inhibition reinstates antigen presentation and T-cell infiltration within PDAC tumors. Notably, we observed no pronounced T cell exhaustion, thus potentially limiting the applicability of PD1/PD-L1-based ICI combinations for sustaining anti-tumor responses. Given the recent advances on antigen-based therapies to improve the tumor antigenicity[54,55], our findings on the combinatory

effect of MEKi and HDACi on supporting the antigen presentation machinery provide functional evidence and avenues to induce more durable immune-mediated tumor elimination in PDAC. Given the notable association between GATA6 and pSTAT1 signaling observed in our study, we speculate that GATA6 might play a central role in IFN-mediated immunoregulation in response to therapeutic intervention. Further investigations are warranted to elucidate the therapy-induced dynamics of GATA6^− centered gene regulatory networks to decipher further, and corroborate its immunomodulatory properties.

## Methods

### Ethics statement
All animal experiments were conducted in accordance with local, national, and European regulations governing the use of laboratory animals. Experimental procedures were approved by the competent animal welfare authorities and performed in compliance with the German Animal Welfare Act (Tierschutzgesetz), the European Directive 2010/63/EU, and the guidelines of the Federation of European Laboratory Animal Science Associations (FELASA).

Animal studies were carried out at Klinikum rechts der Isar, Technical University of Munich, and University Hospital Essen. Experiments performed in Munich were approved from Regierung von Oberbayern, license number was 55.2-1-54-2532-126-2014. Experiments performed in Essen were approved by the Landesamt für Verbraucherschutz und Ernährung (LAVE) Nordrhein-Westfalen. License number for orthotopic mouse models was 81-02.04.2020.A316. License number for patient-derived xenografts models was 84-02.04.2017.A054. License number for CKP was 84-02.04.2017.A315.

According to the approved protocols, the maximal permitted tumor burden was defined as a tumor volume not exceeding 1000 mm³ or 20% of the body weight. Tumor growth was monitored regularly by ultrasound or MRI, depending on the model. Humane endpoints were predefined, and mice were euthanized when tumors approached the maximal allowed size, or earlier if animals exhibited signs of distress, ascites, or body weight loss exceeding 20%.

Due to the interval-based nature of imaging-based tumor monitoring, in rare cases, tumors were found to have exceeded the predefined maximal volume at the final measurement. In these instances, animals were euthanized immediately upon detection, and no procedures were performed beyond the approved humane endpoints. Overall, the study was conducted in strict accordance with the approved animal

The use of human PDAC patient samples in this study was approved by the relevant institutional research ethics boards. For the University Health Network (UHN) cohort, resected treatment-naive stage I/II PDAC specimens were obtained through the UHN Biospecimens Program, with approval from the University Health Network Research Ethics Board (approval no. 17-6106). The Essen cohort

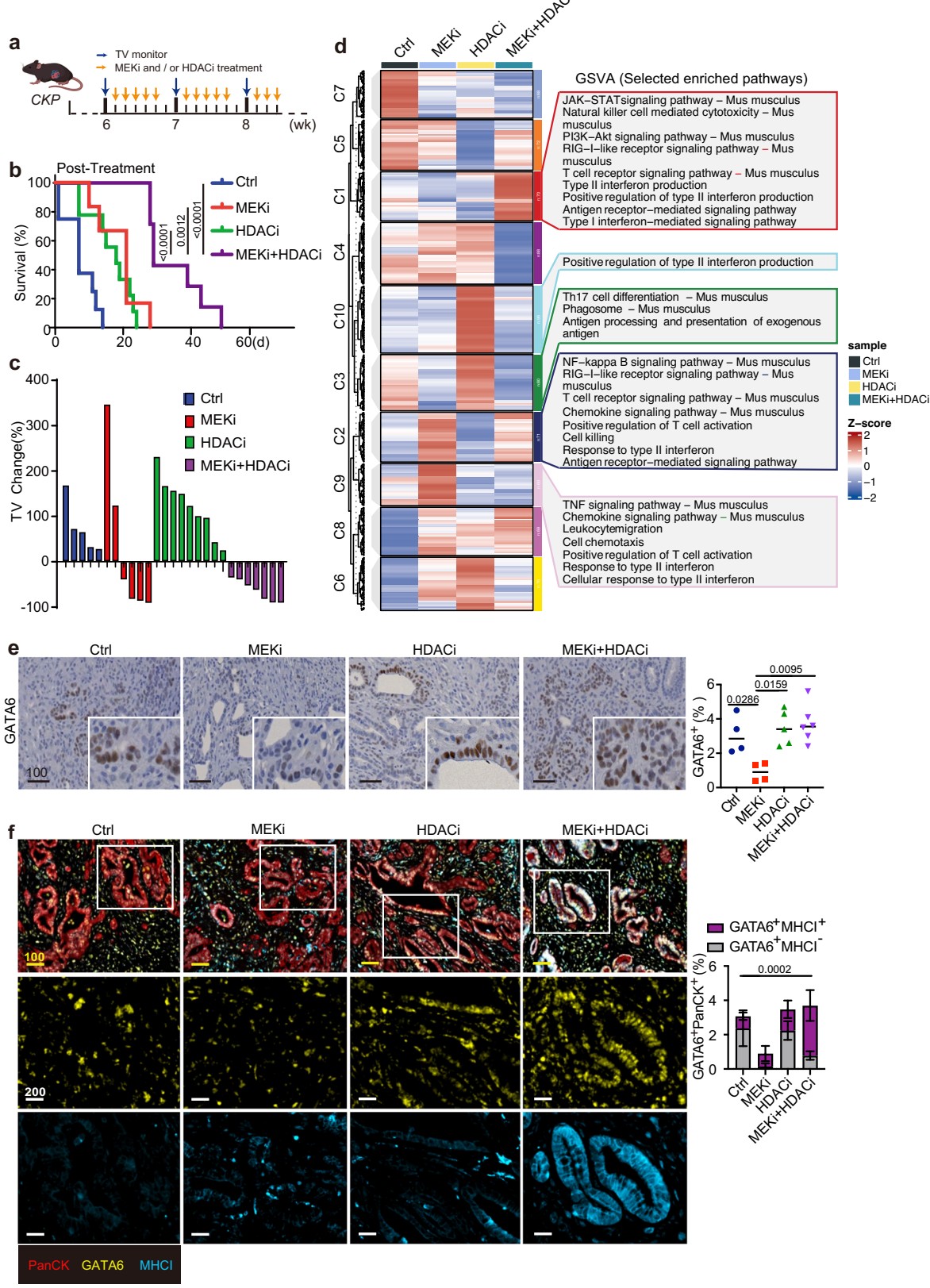

represents a retrospective study conducted in accordance with the recommendations of the local ethics committee of the Medical Faculty of the University of Duisburg-Essen (approval no. 17-7340-BO). Clinical data were retrieved from archived records and electronic health records in compliance with institutional and ethical regulations. For multiplexed spatial imaging analyses, additional PDAC tissue microarray cores were obtained from the Institute of Pathology, Technical University of Munich, under approval from the local ethics committee (ethical vote: 403/17 S).

Written informed consent was obtained from all patients, and the study was conducted in accordance with all relevant local and international ethical guidelines and regulations.

**Fig. 6 | Combined MEK and HDAC inhibition prolongs survival of *CKP* mice with restored GATA6 and MHCI expression. a** Timeline for treatment of *CKP* mice with vehicle control, MEKi, and/or HDACi. Orange arrows indicate the treatment schedule. Blue arrows indicate the schedule of tumor volume (TV) monitoring by 3D ultrasound imaging. Created with BioRender.com. **b** Kaplan-Meier plot shows post-treatment survival analysis from the data of *CKP* mice. Ctrl: $n = 8$ independent mice, median = 7 days; MEKi: $n = 6$ independent mice, median = 21 days; HDACi: $n = 9$ independent mice, median = 18 days; MEKi+HDACi: $n = 7$ independent mice, median = 29 days. Statistical significance of survival differences was determined by the two-sided log-rank (Mantel–Cox) test. **c** Waterfall plot of the tumor volume (TV) percentage change for individual *CKP* mouse in different treatments ($n = 5$ independent mice for Ctrl group, $n = 7$ independent mice for MEKi group, $n = 6$ independent mice for HDACi group, $n = 5$ independent mice for MEKi+HDACi group). **d** Gene set variation analysis (GSVA) of the immune-related transcriptomic profiles of *CKP* tumors treated with or without MEKi and/or HDACi using the NanoString PanCancer Immune Profiling. Left: heatmap showing differential expression of immune-related genes in the 4 treatment groups. Selected statistically significant pathways ($p < 0.05$) enriched in different groups are shown in the right panel. Z-scores represent relative enrichment levels across samples (red: upregulated, blue: downregulated). **e** IHC staining of GATA6 in *CKP* tumors with or without MEKi and/or HDACi treatment. The right panels show the percentage of respective positive cells out of total cells in the whole tumorous tissues as quantified by Definiens software ($n = 4$ independent mice for Ctrl group, $n = 4$ independent mice for MEKi group, $n = 5$ independent mice for HDACi group, $n = 6$ independent mice for MEKi+HDACi group). Mean ± SD is shown. One-way ANOVA, Kruskal–Wallis test were used. **f** mIF staining of GATA6, MHCI, and PanCK was performed in *CKP* tumors. Left panel: representative images showing co-expression of MHCI and GATA6 in PanCK⁺ tumor cells from 4 treatment groups. Right panel: The percentage of GATA6⁺MHCI⁺PanCK⁺ and GATA6⁺MHCI⁻PanCK⁺ cells out of total cells was quantified by HALO software ($n = 4$ independent mice for Ctrl group, $n = 4$ independent mice for MEKi group, $n = 5$ independent mice for HDACi group, $n = 6$ independent mice for MEKi+HDACi group). Mean ± SD is shown. One-way ANOVA and Kruskal–Wallis test were used. Ctrl: vehicle control; MEKi: trametinib; HDACi: domatinostat; MEKi + HDACi: trametinib + domatinostat. Scale bar: μm.

## Cohort description

Immunohistochemical staining of GATA6 and stromal/immune markers was performed in a previously described[15] tissue microarray (TMA) from PDAC patient samples. In brief, the TMA includes 143 resectable tumor specimens from treatment-naive patients with stage I/II PDAC. The information of the patient cohort, including age and sex of subjects, has been published[29]. Resectable tumors were obtained from the UHN Biospecimens Program. Patient samples were mostly accrued at Princess Margaret Cancer Centre at the University Health Network (Toronto, Canada) as reported previously[29].

Multiplexed histological staining for spatial interaction analysis of GATA6⁺ tumor cells, and also Phenocycler analysis, was performed in the "Essen cohort," which is a retrospective study that was carried out according to the recommendations of the local ethics committee of the Medical Faculty of the University of Duisburg-Essen. Clinical data were obtained from archives and electronic health records. In this exploratory retrospective study, a cohort of 54 patients that had undergone pancreatic resection with a final histopathologic diagnosis of human PDAC between March 2006 and February 2016 was used.

6 cases of PDAC primary tumors (four obtained from male sex patients and two from female sex patients), analyzed by the highly multiplex spatial imaging with Phenocycler, were available as TMA cores. Here, we used a tissue microarray compiled at the Institute of Pathology, Technical University of Munich, consisting of primary resected PDAC, resected between April 2008 and May 2020. The clinicopathological data of the patients are shown in Supplementary Table 1.

## Public datasets and bioinformatic analysis

Bulk RNA-sequencing dataset (GSE93326)[30] and single-cell RNA-sequencing dataset (GSE212966)[56] were used to analyze the association of GATA6 expression in tumor with TME. GSE93326 dataset contains 65 pairs of tumor epithelium and stroma LCM samples. "DESeq2" R package was used to identify the Differential Expression Genes (DEGs) between GATA6^high and GATA6_low samples. GSE212966 consists of 6 PDAC and 6 adjacent noncancerous resection specimens. We used the 6 PDAC samples data to do analysis. "Seurat" R package was used to standardize the expression of filtered samples and identify the top 2000 genes with the most noticeable difference between cells. Cells were filtered out with the threshold of the ratio of mitochondrial genes ≤15%. Genes expressed in >3 cells and cells with at least 200 genes were retained. An integrated dataset was created by using "Harmony" R package. The integrated data then proceeded with principal component analysis (PCA), and the dimensions of the top 30 PCs were reduced by the tSNE and UMAP algorithm to obtain principal clusters. FindAllMarkers function in "Seurat" R package was used to identify marker genes of each cluster. Finally, the cells were annotated according to marker genes expression. DEGs between GATA6⁺ tumor cells and GATA6⁻ tumor cells were analyzed using the Wilcoxon rank sum test with Bonferroni correction included in FindMarkers function in "Seurat" R package. "CellChat" R package and CellChatDB database were used to infer the cellular interactions among GATA6^+/- tumor cells and CD20⁺ B cells, CD8⁺ Tcells, CD4⁺ T cells, and FOXP3⁺ T cells by using the expression of known ligand-receptor pairs and identifying the changes in cellular interaction, the netVisual_circle function was utilized for evaluating the strength of cellular interaction networks among cell types. Gene Set Enrichment Analysis (GSEA) was conducted using the GSEA function from the "ClusterProfiler" R package. The gene chip annotation file Human Gene Symbol with_Remapping_MSigDB.v7.4.chip was utilized, and genes were collapsed and remapped to gene symbols. The analysis incorporated gene sets from Hallmarks (h.all.v7.5.1), KEGG (2.cp.kegg.v7.5.1), and GO (c5.go.v7.5.1).

## Phenocycler multiplex tissue staining and imaging

Formalin-fixed paraffin-embedded (FFPE) tissue was cut at 5 μm thickness and mounted onto poly-lysine-coated coverslip (for resected tumors); while for TMA cases, FFPE tissue was cut at 5 μm thickness and mounted onto microscope slides (Superfrost®). The tissues were deparaffinized and rehydrated, and heat-induced epitope retrieval was performed at 97 °C for 10 min. The tissue was stained with an antibody cocktail with 27 antibodies or 40 antibodies to a volume of 100 μl overnight at 4 °C in a sealed humidity chamber on a shaker. After multiple fixation steps using 1.6% paraformaldehyde, 100% methanol, and BS3 (Thermo Fisher Scientific, #21580), the coverslip was mounted onto a custom-made acrylic plate (Bayview Plastic Solutions). Imaging was performed with a Keyence BZ-X710 inverted fluorescence microscope, or PhenoImager FUSION microscope (Akoya Biosciences). For resected tumor tissues imaged by Keyence, four regions of interest (ROIs) were selected from the resected tissues for analysis; while for TMA imaged by FUSION, the whole section of all tissue cores were imaged. Antibody information, light exposure times and the arrangement of cycles are shown in Supplementary Dataset 6.

## Data processing of CODEX and Phenocycler/FUSION images

Raw TIFF image files were processed using the CODEX Toolkit (github.com/nolanlab/CODEX). After processing, the staining quality for each antibody was visually assessed in each tissue microarray spot, and cell segmentation was performed using the DAPI nuclear stain. Marker expression was quantified, and single-cell data were saved as FCS files, which were then imported into CellEngine (cellengine.com) for cleanup gating. After that, the data was analyzed by HALO for

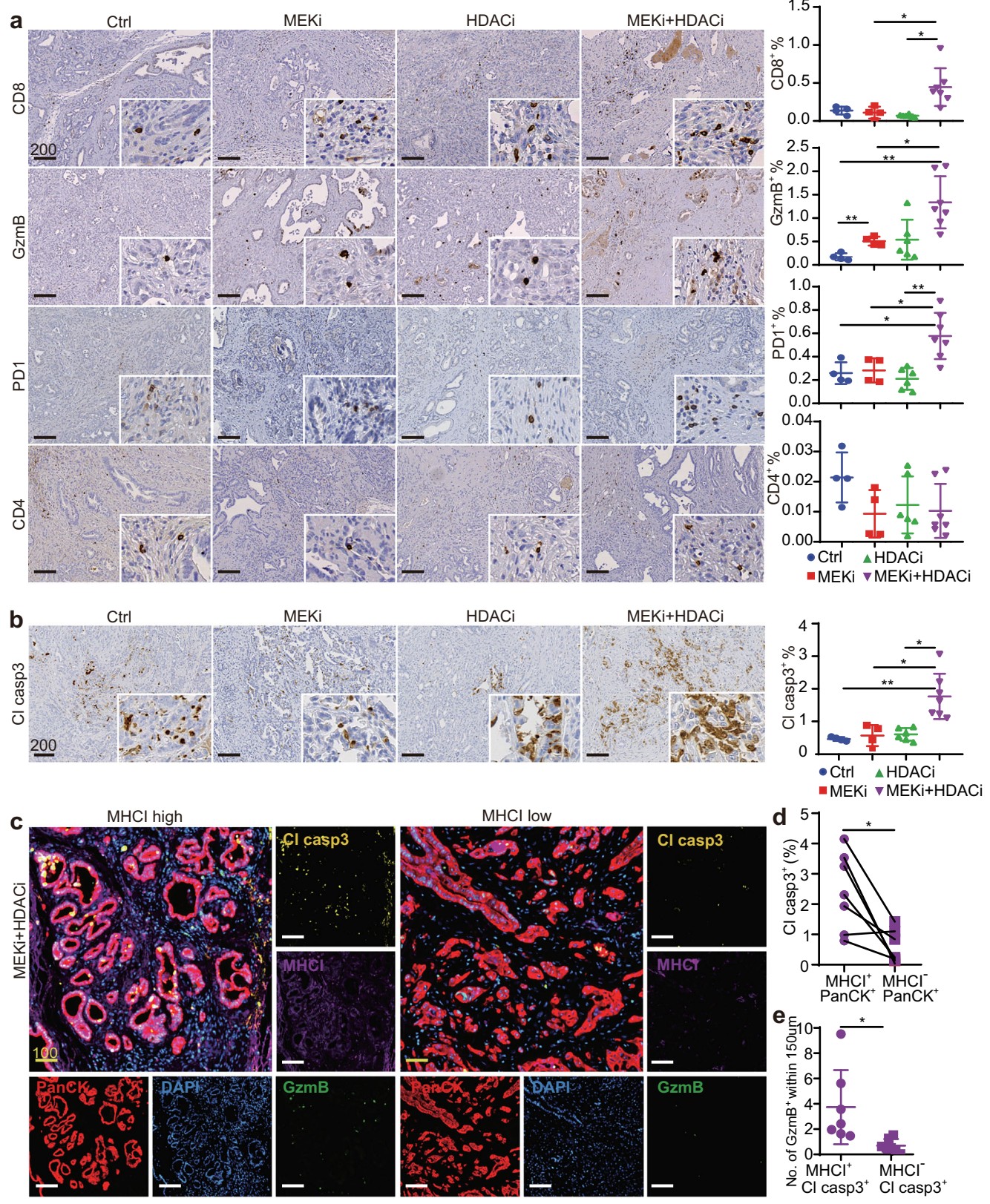

quantification and spatial interaction analysis. For CODEX, four ROIs were selected from each tumor tissue. Average of the four ROIs was shown and represent one patient.

## Multiplexed immunofluorescent (IF) histological staining
Multiplexed IF was conducted using the Opal multiplex system (Akoya Biosciences, MA) in accordance with manufacturer's instructions. In

brief, FFPE sections were deparaffinized, then fixed with 4% paraformaldehyde, and the antigens were retrieved using tris/EDTA (pH9) to induce epitope retrieval by heat-induced epitope retrieval. Each section was put through multiple rounds of staining; each included endogenous peroxidase blocking and protein blocking, as well as primary antibody and the corresponding secondary horseradish peroxidase-conjugated polymer (Zytomed Systems, Germany, or

**Fig. 7 | Combined MEKi and HDACi treatment enhances cytotoxic T cell infiltration and tumor apoptosis in vivo.** IHC staining of **a** T cell marker CD8 and CD4, activated T cell marker PD1, cytotoxic T cell marker GzmB, and **b** apoptosis marker Cl casp3 in *CKP* tumors with designed treatments. The right panels show the percentage of respective positive cells out of the total cells in the whole tumorous tissues as quantified by Definiens software (*n* = 4 independent mice for Ctrl group, *n* = 4 independent mice for MEKi group, *n* = 6 independent mice for HDACi group, *n* = 7 independent mice for MEKi+HDACi group). Mean ± SD is shown. One-way ANOVA and Kruskal−Wallis test were used. **c** mIF staining of combinatory MEKi and HDACi-treated *CKP* mice showing increased GzmB$^+$ and Cl casp3$^+$ cells in high

tumor MHCI$^+$ expression areas (*n* = 7). **d** Co-expression analysis shows the percentage of Cl casp3$^+$ tumor cells within MHCI$^+$PanCK$^+$ and MHCI$^-$PanCK$^+$ population (*n* = 7). The percentage of respective positive cells out of the total cells was quantified by HALO software. Statistical significance was calculated by two-tailed Wilcoxon matched-pairs signed rank test. **e** Computational spatial analysis shows higher number of GzmB$^+$ cells in close proximity (<150 μM radical distance) of MHCI$^+$ Cl casp3$^+$ cells when compared to MHCI$^-$ Cl casp3$^+$ cells (*n* = 7). Quantified by Halo software. Mean ± SD is shown. Statistical significance was calculated by unpaired two-tailed Mann−Whitney test. Ctrl: vehicle control; MEKi: refametinib; HDACi: domatinostat; MEKi+HDACi: refametinib + domatinostat. Scale bar: μm.

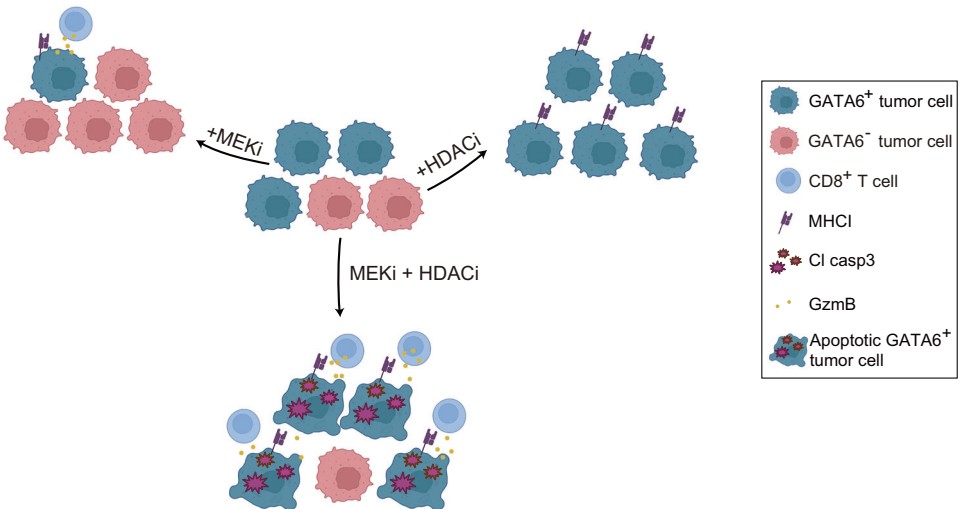

**Fig. 8 | GATA6-dependent regulation of tumor MHCI expression and anti-tumor cytotoxicity.** Schematic diagram illustrating how GATA6-dependent MHCI expression in tumor cells is associated with enhanced antitumor cytotoxicity following combined MEK inhibitor (MEKi) and HDAC inhibitor (HDACi) treatment.

Akoya Biosciences). Tyramide signal amplification was used to bind each horseradish peroxidase-conjugated polymer with different fluorophores. To remove antibodies before the next round of staining, additional antigen retrieval in heated Tris/EDTA (pH9) was performed. Sections were counterstained with DAPI (Vector lab) after all sequential staining reactions. The sequential multiplexed staining protocol is shown in Supplementary Table 2. With 10x objective magnification, slides were scanned and digitalized by Zeiss Axio Scan.Z1, ZEN Blue software version 3.1 (Carl Zeiss AG, Germany). Quantification of individual and/or co-expressing markers, and spatial interaction analysis in the mIF images was performed with HALO version 3.2 (Indica Labs).

### Genetic mouse strains and models
The study was not applied to only one gender. Mice (both sexes, aged 6–12 weeks) were randomly assigned to experimental groups. No specific sex or age selection was made. *Ptf1a$^{wt/Cre}$;Kras$^{wt/LSL-G12D}$;Trp53$^{tm1.1Dgk}$* (*CKP*) mice spontaneously develop PDAC at 6 weeks[40,57]. *FKPC2GP* mice were generated by crossing *Ptf1a$^{wt/Flp}$;Kras$^{wt/FSF-G12D}$;Trp53$^{tm1.1Dgk}$* (*FKP*) mice to *Gt(ROSA)26Sor$^{tm3(CAG-Cre/ERT2)Dsa}$* (*R26$^{FSF-CAG-CreERT2}$*) and *Gt(ROSA)26Sor$^{tmloxP-STOP-loxP-GP-IRES-YFP}$* (*R26$^{LSL-GP}$*) strains[39–41,58]. For treatment experiments, mice were randomized. None of the mice with the appropriate genotype were excluded from this study.

### Establishment of orthotopic mouse models
For the generation of orthotopic PDAC tumors in mouse based on *CKP*-derived cell lines (60590 and 110299) and AID system cells (110299$^{AID-GATA6}$ and 110299$^{WT}$), 5 × 10$^3$ cells were resuspended in 30 μL of a 1:1 dilution of Matrigel® (Corning) and cold plain medium and injected into the pancreas of C57Bl/6J (B6J) mice using insulin syringes (BD micro-fine 30 Gauge) under the guidance of ultrasonic imaging (visual

sonic vevo 2100, FUJIFILM Sonosite, Canada). The injection was considered successful by the circular appearance of encapsulated cell suspension without signs of leakage. After orthotopic transplantation, tumor volume of tumor-bearing mice was monitored by 3D mode of ultrasound imaging once a week[59].

### Establishment of patient-derived xenografts
Fresh tumor fragments were transplanted subcutaneously (s.c.) into the left flank of anaesthetized NOD scid gamma mice. Mice were maintained under sterile and controlled conditions (22 °C, 50% relative humidity, 12 h light−dark cycle, autoclaved food and bedding, acidified drinking water). Tumor growth was measured twice weekly, and tumors were routinely passaged at TV = 1000 mm³. Xenograft material was snap frozen and stored at −80 °C or processed to FFPE blocks.

### In vivo treatment of mice
For all orthotopic PDAC models, treatment was initiated when tumors reached a volume of 200–400 mm³. Mice were treated with either vehicle control (2% DMSO + 40% PEG300 + 5% Tween-80 + 53% ddH$_2$O), refametinib (RDEA119, BAY 86-9766, 25 mg/kg), domatinostat (4SC-202, 40 mg/kg), or a combination of refametinib and domatinostat. Drugs were administered by oral once daily for 5 consecutive days, followed by a 2-day treatment break, in repeated cycles. For mice bearing orthotopic tumors derived from the AID-GATA6 system cell lines, 5-Ph-IAA (auxin, 20 mg/kg) was administered intraperitoneally (i.p) once daily, starting 3 days before the initiation of refametinib treatment and continuing throughout the treatment period. For CD8$^+$ T cell depletion experiments, anti-CD8 antibody (100 μg/mouse, i.p.) was given once daily for 2 consecutive days, followed by a 3-day rest period. Afterward, combination therapy with refametinib and

domatinostat was initiated as described above. Tumor volume in orthotopic models was monitored weekly using ultrasound starting 2 weeks after tumor cell implantation.

For PDX models, treatment was initiated when tumor volume reached approximately 200 mm³. Mice received either vehicle control (2% DMSO + 40% PEG300 + 5% Tween-80 + 53% ddH₂0), trametinib (GSK1120212, 1 mg/kg), mocetinostat (MG0103, 40 mg/kg), or the combination of trametinib and mocetinostat. Drugs were administered by oral once daily for 5 consecutive days, followed by a 2-day treatment break, in repeated cycles for 4 weeks.

For *CKP* mice, treatment started at 6 weeks of age when the tumor diameter of the tumor was approximately 5 mm with vehicle control, trametinib, domatinostat, or the combination of trametinib and domatinostat. Drugs were administered the same as PDX models. MRI was performed according to relevant guidelines and regulations[60,61]. Tumor burden was measured by ultrasound weekly from 5 weeks old. Mice were euthanized when tumors reached a volume >1000 mm³, exhibited >10% body weight loss, developed ascites, or showed signs of distress. Tumors were harvested for further experiments.

## Immune mRNA profiling

All procedures related to mRNA quantification, including sample preparation, hybridization, detection, and scanning, were carried out by NanoString Technologies for PanCancer Immune Profiling codesets (NanoString Technologies). Normalized data were log₂-transformed and then used as input for differential expression clustering and for heat maps. For the PanCancer Immune codeset, genes previously shown to be characteristic of various immune cell populations were used to measure the abundance of these populations with adjusted *p* value < 0.05, Benjamini-Yekutieli false discovery rate (FDR) < 0.1. nSolver Software version 4 (Nanostring Technologies) was used for analysis. Subsequent GSVA performed with R Studio with gene expression data after processing with nSolver.

## Immunohistochemistry

All IHC experiments were conducted on FFPE sections. For IHC staining of GATA6, CD3, CD8, CD20, CD4, CD68, and FoxP3 on a TMA of *n* = 143 human PDA tissues, digital quantification by QuPath[62] and data processing within our in-hour pipeline ("HOURGLASS") were reported previously[29,63]. In brief, GATA6 expression was quantified within malignant epithelia, based on manual annotations drawn on every core by a trained pathologist. CD3⁺ and CD8⁺ TILs were manually counted by a trained pathologist and normalized by total tissue area. CD4⁺ and FoxP3⁺ cells were digitally quantified and normalized by total tissue area. CD20⁺ and FoxP3⁺ cells were digitally quantified and normalized to the CK19⁻ area fraction of the tissue area, since these populations maintained a near exclusive localization to stromal regions.

For all the remaining IHC stainings performed in the present study, heat-induced epitope retrieval was performed with Tris/EDTA (pH9). A serum-free protein blocking solution (Zytomed System) was used to block the slides, followed by incubation with primary antibodies (Supplementary Table 3) for 1 h at room temperature, and secondary antibodies for 30 min at room temperature, and then DAB chromogen development was performed. As for the staining of H-2Db, a mouse-on-mouse (MOM) immunodetection kit (BMK-2202, Vector Laboratories) was used. Counter-staining with hematoxylin, dehydrating, and mounting the slides followed.

With a 10× objective magnification, the slides were scanned and digitalized by Zeiss Axio Scanner Z.1 (Carl Zeiss AG, Germany). A quantitative analysis of the percentage of positive IHC-stained cells was conducted by Definiens version 2.6 (Definiens AG, Germany). Nuclei staining (hematoxylin) detected by the software was used to calculate the number of cells in the entire tissue section.

## Cell culture and treatment

Primary murine PDAC cell lines and GP cell lines[5,39] had been derived previously from corresponding tumor pieces of *CKP* and *FKPC2GP* mice through incubation in DMEM high-glucose medium (Thermo Fisher Scientific) supplemented with 10% fetal bovine serum (Thermo Fisher Scientific) until proliferating tumor cells could be established. 2838c3 and 6694C2 *KPCY* cell lines were purchased from Kerafast (CAT#EUP013-FP, Boston, USA). The long-term trametinib-treated murine PDAC cell lines were established by treatment with increasing doses of trametinib (GSK1120212) until 100x of the cells' initial IC50 and maintained in 10% FBS-supplemented DMEM with corresponding doses of trametinib.

*CKP* cells, both parental and long-term trametinib-treated, were seeded in 6-well plates (200K/well) and adhered 8 h. Cells were then treated with DMSO (concentration-matched vehicle control), or trametinib (78.125 nmol/L), or domatinostat (312.5 nmol/L), or the combination of trametinib (78.125 nmol/L) and domatinostat (312.5 nmol/L) for 72 h.

For auxin-induced degradation, *CKP* cells (110299) were incubated with or without 1 μM 5-Ph-IAA (Bio Academia, Japan) in culture medium for different time points.

## BrdU Incorporation Assay

Cell proliferation was measured using the BrdU Cell Proliferation Assay Kit (Sigma-Aldrich, QIA58) according to the manufacturer's instructions. Briefly, $1 \times 10^5$ cells/mL were seeded in 96-well plates (100 μL/well) and allowed to attach for 24 h. BrdU (1:2000) was added (20 μL/well), and cells were incubated for another 24 h at 37 °C. Blank (medium only) and background (cells without BrdU) controls were included. After fixation and denaturation, wells were incubated sequentially with anti-BrdU antibody (1:100, 1 h), HRP-conjugated secondary antibody (1:1000, 30 min), and substrate solution (15 min, dark). After adding stop solution, absorbance was measured at 450 nm with a reference of 595 nm using a Tecan plate reader.

## Generation of GATA6 knockout cell lines

Modified from lentiCRISPR v2 (Plasmid #52961, Addgene), pLentiCRISPR-V2.1-GFP contains inserts Cas9, sgRNA scaffold, GFP, and Puromycin resistance. After inserted with sgRNA targeting Gata6 (sgGata6) or nontargeting control sequence (sgNT), pLentiCRISPR-V2.1-GFP-sgGata6 were used for CRISPR-Cas9 genome editing of *KPCY* 2838c3 cells. Modified from lentiGuide-Puro (Plasmid #52963, Addgene), plentiGuide-Puro-BFP contains the inserts sgRNA scaffold, BFP, and Puromycin resistance. After inserted with sgRNA targeting Cas9 (sgCas9), plentiGuide-Puro-BFP-sgCas9 was used for Cas9 removal. Modified from pCDH-EF1 (Plasmid #72266, Addgene), pCDH-EF1a-BFP-GP33-GP41 contains inserts BFP and GP33-GP41 peptide, which is H-2Db-restricted epitope generated from the lymphocytic choriomeningitis virus (LCMV). The gRNA sequences are as follows: sgCas9-1CAGATAGATCAGCCGCAGGT; sgGATA6_1 TACGTGCCCACCACGCGCGT; sgGATA6_2CCGGGAGTGGAGCTCCCGCG; sgGATA6_3 AGGGCGAGTAGGTCGGGTGA; sgGATA6_4 TCCGCCGACAGCCCCCGTA. Specifically, 2838c3 cell lines were first transduced with pLentiCRISPR-V2.1-GFP-sgRNA targeting Gata6 (sgGata6) or non-targeting control sequence (sgNT) and then transduced with plentiGuide-Puro-BFP-sgRNA targeting Cas9 (sgCas9) 7 days after first transduction, followed by sorting of GFP and BFP double-positive cells. After a 7-day recovery, the cells are ready for the following functionality experiments. To express GP33-41, 2838c3 cell lines were transduced with pCDH-EF1a-BFP-GP33-GP41, followed by sorting of BFP-positive cells and 4 days' recovery. The cells are then ready for the following functionality experiments.

## Western blot analysis

Cells or tissues were lysed in RIPA buffer (Cell Signaling Technology, 9806) supplemented with protease inhibitor (Sigma-Aldrich, 4693124001) and phosphatase inhibitor (Sigma-Aldrich, 4906837001). Lysates were incubated on ice for 30 min and then centrifuged at $12,000 \times g$ for 15 min at 4 °C. Protein concentration was determined using the Pierce BCA Protein Assay Kit (Thermo Fisher, 23225). Equal amounts of protein (20 μg) were loaded onto SDS-PAGE gels (8–12%) and transferred to PVDF membranes. After blocking with 5% BSA in TBST for 1 h at room temperature, membranes were incubated overnight at 4 °C with primary antibodies: anti-GATA6 (Abcam, ab175349, 1:1000), anti-GAPDH (Cell Signaling Technology, 9098, 1:1000), anti-pSTAT3 (Cell Signaling Technology, 9145S, 1:1000), Vinculin (Cell Signaling Technologies,13901S, 1:1000), β-actin (Abcam, ab8227, 1:1000), ERK1/2 (Cell Signaling Technology, 4695 T, 1:1000), phospho-ERK1/2 (pT202/pT204, Cell Signaling Technology, 4376S, 1:1000), E-Cadherin (Cell Signaling Technology, 3195S, 1:1000), and N-Cadherin (Abcam, ab76057, 1:1000). HRP-conjugated secondary antibodies were applied for 1 h at room temperature. Blots were developed using WesternBright ECL HRP Substrate (Advansta, K-12045-D20), followed by imaging with a chemiluminescent imaging system.

## Endogenous knock-in of AID tag

A homology-directed repair (HDR) template containing homology arm (HA) homologs to the sequences upstream and downstream of the gata6 start codon was cloned. HAs were PCR-amplified using murine primary PDAC cell line (110299) genomic DNA as template and cloned into pJET (Thermo Fisher Scientific) flanking the Blast-P2A-V5-AID cassette to generate pJET_gata6_N-AID_HDR. Two sgRNAs targeting around the gata6 start codon were cloned into PX458 (Addgene #48138) to obtain PX458_gata6_sgRNA1 and PX458_ gata6_sgRNA2. The PAM sites of the two sgRNAs were mutated in the HDR using the primers for the HA PCR amplification.

To obtain stable cell lines expressing endogenous GATA6 with an AID tag at the N-terminal, Murine primary PDAC cells (110299) were grown in 6-well dishes and transfected with 3 μg of each PX458_gata6_sgRNA1 or PX458_gata6_sgRNA2 and pJET_gata6_N-AID_HDR plasmids using polyethylenimine (PEI). After 72 h of transfection, cells were trypsinized, reseeded into several 15 cm dishes with various dilution and selected with 15 μg/ml Blasticidin. After 7 days of selection, colonies were picked and transferred to 24-well plates. Individual clones were evaluated using genomic PCR and western blot. PCR products from some clones were purified using the GeneJET Gel Extraction kit (Thermo Fisher Scientific) and sent for Sanger sequencing (LGC Genomics). For TIR1[F74G] expression in 110299[AID-GATA6] cells, TIR1[F74G] and eGFP were cloned into pRRLSin.cPPT.SFFV-IRES-Hygro vector to obtain pRRL-hygro-TIR1(F74G)-T2A-eGFP. 110299[AID-GATA6] cells were then lentivirally transduced with pRRL-hygro-TIR1(F74G)-T2A-eGFP vector and selected with 500 μg/ml hygromycin (InvivoGen) for a week.

## RNA sequencing data

Generated as described[35], available in the GEO database under accession code GEO: GSE146348.

## Hierarchical clustering

Hierarchical clustering of parental vs. long-term trametinib-treated *CKP* cells based on PDAssigner genes or PDAC subtype-associated genes defined by Bailey et. al.[44] and Moffitt et al.[64]. Benjamini-Hochberg adjusted *p* value < 0.01; log₂ fold change > 1 or log₂ fold change < −1.

## Gene set enrichment analysis (GSEA)

GSEA was performed using default settings of Broad Institute algorithm, and gene set permutation on the Hallmarks and KEGG gene sets. Benjamini-Hochberg FDR *q* value < 0.05, NES > ± 1.5.

## Flow cytometric analysis

For cell surface MHCI and PD1 expression, cells were washed with washing buffer (2% FBS in PBS) and then incubated with antibodies: MHCI (Biolegend, 111513, lot: B351983); PD1 (Biolegend, 135213), or an equal amount of corresponding isotype control. For intracellular TNF, GzmB, and GATA6 expression, cells were fixed with 4% paraformaldehyde for 10 min at 37 °C. After washing twice with washing buffer, cells were permeabilized with 0.1% Saponin for 20 min and then stained with antibodies: TNFa (Biolegend, 506306, lot: B290019); GzmB (BioLegend, 515406, lot: B301362); GATA6 (Cell Signaling Technology, 26452S, lot: 4) and corresponding isotype. Cells were then washed, resuspended, and subjected to analysis. Expression of corresponding molecules of 10,000 viable cells was analyzed by flow cytometry (FACSCelasta, FACSDiva software, version 5.0; BD Biosciences) as mean fluorescence intensity. Raw data were analyzed using FlowJo software version 7.5.5 (Tree Star Inc., Ashland, OR).

## Isolation of T cells from *P14-TCR-Tg* mice

Spleens were collected from *P14-TCR-Tg* mice. Afterwards, they were homogenized freshly, then lysed in Ammonium-Chloride-Potassium lysis buffer. In accordance with manufacturer's instructions, T cells were negatively selected by MACS (Miltenyi Biotec). After isolation, T cells were labeled with 5 μM CFSE (Thermofisher) and resuspended in 10% FBS-supplemented RPMI medium in prestimulated with 20 ng/ml IL-2 (Peprotech) for 1 h, before co-culture with GP cell lines.

## Co-culture of LCMV-gp33-reactive T cells and GP cells

After treatment with or without tamoxifen to induce LCMV-gp33 expression for 2 days, GP cells were treated with the same four groups of treatments for *CKP* cells for 72 h. Following that, the cells were harvested and seeded in 6-well plates containing 10% FBS-supplemented DMEM and were ready for co-culture. LCMV-gp33-reactive T cells were subsequently isolated and then cultured in DMEM supplemented with 10% FBS for 48 h with or without GP cell lines at a ratio of 8:1. The cells were then photographed under a microscope, then harvested for propidium iodide (PI) staining to determine whether they have anti-tumor cytotoxicity. Furthermore, GzmB, TNFa, and PD1 were stained to assess the activity of T-cells.

## Statistical analysis

All statistics were performed using GraphPad Prism version 8.0 (GraphPad Software, La Jolla, CA) or R software (version 4.1.3). Survival data were analyzed by two-sided log-rank (Mantel–Cox) test, while correlation analysis was by Spearman's rank correlation coefficient. The Chi-square test was used to assess the independence between two categorical variables. Unpaired two-tailed Mann–Whitney test was applied for non-normally distributed data comparison between unpaired two groups. For paired two groups, two-tailed Wilcoxon matched-pairs signed rank test was used. For multiple group comparison, One-way ANOVA and Kruskal–Wallis test were used. Data are represented as mean ± S.D. The exact *p* values are shown in the figures.

## Reporting summary

Further information on research design is available in the Nature Portfolio Reporting Summary linked to this article.

# Data availability

Publicly available RNA sequencing datasets used in this study were obtained from the Gene Expression Omnibus (GEO) database under accession numbers: GSE146348, GSE93326, GSE212966. These datasets were generated by independent studies and were not produced as part of this publication. Source data are provided with this paper.

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

## Acknowledgements

J.T.S. is supported by the German Cancer Consortium (DKTK), by the Deutsche Forschungsgemeinschaft (DFG, German Research Foundation) through 405344257 (SI 1549/3-2), 421166016 (SI1549/4-1), and 450917483 (GRK2762/1), by the German Federal Ministry of Education and Research (BMBF; 01KD2206A/SATURN3), the European Union's Seventh Framework Programme for research, technological development and demonstration (FP7/CAM-PaC) under grant agreement no. 602783, by the Deutsche Krebshilfe (DKH, German Cancer Aid) through 70115201 (TACTIC) and the research network CANcer TARgeting (CANTAR) of the Ministry of Culture and Science of the State of North Rhine-Westphalia (MKW NRW). Multiplex imaging analysis by Phenocycler was performed at the Imaging Center Essen (IMCES), a Service Core Facility of the Faculty of Medicine of the University Duisburg-Essen, Germany. This work was partly funded by the German Research Foundation (DFG, Deutsche Forschungsgemeinschaft) Project-ID 418179183 (KFO 337): RO 3577/7-1, RO 3577/8-1, SI 1549/3-1, SI 1549/3-2 (J.T.S.). This work was supported by grants from the German Research Foundation (DFG, WO 2108/2-1, TRR387), the German Cancer Aid (funding of TACTIC), and the European Research Council (PROTAC-PDAC: #101087045) to E.W. The authors would like to thank Drs. Anton Berns, Howard Crawford, David Kirsch, Jos Jonkers, Tyler Jacks, Doron Merkler, Hassan Nakhei, Dieter Saur, Roland M. Schmid, David Tuveson, for providing transgenic animals. The authors acknowledge support by the Westdeutsche Biobank Essen (WBE) and IBioTUM tissue Munich. 4SC provided domatinostat. There was no involvement of funding sources in the design of the study; data collection, analysis, and interpretation; or publication decision.

## Author contributions

Concept and design: J.P., J.Y., P.F.C., and J.T.S. Acquisition of data: B.G., R.B., R.L., D.B., S.L., B.T.G., N.K., K.S., C.M., P.F.C., J.T.S., P.F.C., J.P., J.Y., G.A., R.F., L.G., B.A., M.V., G.G., S.D., and X.W. performed in vitro experiments; J.P., J.Y., K.A., M.T.A., N.T., and A.G. performed in vivo experiments and analyzed data; J.P., J.Y., G.A., R.F., G.G., A.S., K.S., and P.F.C. performed histological stainings and analysis. Analysis and interpretation of data: J.P., L.G., and S.L. performed bioinformatic analysis and assisted data interpretation; K.L. assisted the data interpretation of LCMV-gp33-expressing genetic PDAC model systems; J.P., J.Y., S.H., E.W., C.S., P.F.C., and J.T.S. interpreted the results of experiments. Writing and revision of manuscript: J.P., J.Y., G.A., P.F.C., and J.T.S. All authors commented on the manuscript and approved the final version. Study supervision: P.F.C. and J.T.S. Funding acquisition was contributed by J.T.S.

## Funding

## Competing interests

J.T.S. receives honoraria as a consultant or for continuing medical education presentations from AstraZeneca, Bayer, Boehringer Ingelheim, Bristol-Myers Squibb, Immunocore, MSD Sharp Dohme, Novartis, Roche/Genentech, and Servier. His institution receives research support from Abalos Therapeutics, AstraZeneca, Boehringer Ingelheim, Bristol-Myers Squibb, Eisbach Bio, Oncolytics Therapeutics and Roche/Genentech; he holds ownership in FAPI Holding (<3%), all outside the submitted work. The remaining authors declare no competing interests.

## Additional information

JuanFei Peng[1,2,3,19], JiaJin Yang[1,2,19], Georgia Antonopoulou [1,2,19], Rui Fang[1,2], Bikash Adhikari [4], Markus Vogt [4], Elmar Wolf [4], Chong Sun [5], Shangce Du[5], Laura Godfrey[1,2], Aayush Gupta[6], Marija Trajkovic-Arsic[1,2], Nicole Teichmann[6], Barbara T. Grünwald[2,7,8], Niklas Krebs[2,7,8], Katja Steiger [9,10], Carolin Mogler [9,10], Kristina Althoff[1,2], Xin Wang[1,2], Giovanni Giglio [1,2], Sven-Thorsten Liffers[1,2], Konstantinos Savvatakis[1,2], Rickmer Braren [10,11], Rita T. Lawlor [12,13], Aldo Scarpa [13,14], Diana Behrens[15], Karl S. Lang[16], Phyllis F. Cheung [1,2,17] ✉ & Jens T. Siveke [1,2,18] ✉

[1]Bridge Institute of Experimental Tumor Therapy (BIT) and Division of Solid Tumor Translational Oncology (DKTK), West German Cancer Center, University Hospital Essen, University of Duisburg-Essen, Essen, Germany. [2]German Cancer Consortium (DKTK), partner site Essen, a partnership between German Cancer Research Center (DKFZ) and University Hospital Essen, Essen, Germany. [3]Department of Gastroenterology, Sun Yat-sen Memorial Hospital, Sun Yat-sen University, Guangzhou, China. [4]Institute of Biochemistry, University of Kiel, Kiel, Germany. [5]Division Immune Regulation in Cancer, German Cancer Research Center (DKFZ) Heidelberg, Heidelberg, Germany. [6]Department of Internal Medicine II, Klinikum rechts der Isar der Technischen Universität München, Munich, Germany. [7]Department of Urology, West German Cancer Center, University Hospital Essen, Essen, Germany. [8]Princess Margaret Cancer Centre, University Health Network, Toronto, ON, Canada. [9]Institute of Pathology, School of Medicine and Health, Technical University of Munich, Munich, Germany. [10]German Cancer Consortium (DKTK), Partner Site Munich, Munich, Germany. [11]School of Medicine and Health, Klinikum rechts der Isar, Technical University of Munich (TUM), Munich, Germany, Department of Diagnostic and Interventional Radiology and Department of Nuclear Medicine, University Medical Center Hamburg Eppendorf, Hamburg, Germany. [12]Department of Engineering for Innovation Medicine, University of Verona, Verona, Italy. [13]ARC-Net Research Centre, University and Hospital Trust of Verona, Verona, Italy. [14]Department of Diagnostics and Public Health, University of Verona, Verona, Italy. [15]EPO - Experimental Pharmacology and Oncology GmbH, Berlin, Germany. [16]Institute of Immunology, Medical Faculty, University of Duisburg-Essen, Essen, Germany. [17]Spatiotemporal tumor heterogeneity, DKTK, partner site Essen, a partnership between DKFZ and University Hospital Essen, Essen, Germany. [18]National Center for Tumor Diseases (NCT) West, Campus Essen, Essen, Germany. [19]These authors contributed equally: JuanFei Peng, JiaJin Yang, Georgia Antonopoulou. ✉e-mail: f.cheung@dkfz-heidelberg.de; j.siveke@dkfz.de

