## [Transparent Peer Review file · Nature Communications]

Combined targeted and epigenetic-based therapy enhances antitumor immunity by stabilizing GATA6-dependent MHC1 expression in pancreatic ductal adenocarcinoma

Corresponding Author: Professor Jens Siveke

Version 0:

Reviewer comments:

Reviewer #1

(Remarks to the Author)

The authors present a manuscript that assesses the ability of MEK inhibitors to alter MHC1 levels depending upon the level of GATA6 in pancreatic cancer cell lines. The present a convincing MFI analysis of two high GAT6 versus two low GAT6 cell lines to demonstrate only the high GATA6 cell lines increase MHC1 when exposed to MEKi. As MHC1 is critical for immune recognition and/or evasion, mechanisms that can increase MHC1 expression are critical for improving immunotherapy. The authors further combine MEKi with HDACi, which shows increased survival of CKP pancreatic cancer bearing mice. The mechanisms behind the HDACi is not well laid out or tested, leaving a GAP in the understanding and applicability of this combination therapy. This is a highly correlative study that provides novel insight into the roles of MEK and GATA6 in pancreatic cancer, yet multiple moderate concerns decrease the enthusiasm for this manuscript.

Concerns:

Writing issues:

- 1.) The introduction and some of the discussion are not well written and lack clarity for multiple statements; they are generally broad and do not provide the reader with accurate/specific interpretations from the cited references. Especially with respect to previous studies on GATA6 in the introduction. Additionally, the First two lines in the abstract are too broad, unsupported, and should be rewritten.
- 2.) Multiple small grammatical issues are detected throughout the manuscript.
- 3.) Methods for ultrasound imaging of the tumors is not provided.
- 4.) GP99 cells should not be portrayed as GATA6- as this implies negative; in reality they do express low levels (fig3d). It is suggested to use the term GATA6lo. It is also suggested that a GATA6 knockout be test as cancer cells are highly plastic and the ability to reexpress GATA6 in lo cells is unclear.
- 5.) GATA6 does not “orchestrate” EMT, instead the reference demonstrates that GATA6 is required to maintain epithelial differentiation. The following is therefore quite misleading and should be rewritten : “Recent seminal studies have illuminated the crucial tumor-suppressive role of GATA6 in orchestrating epithelial-to-mesenchymal transition (EMT) and restraining tumor dissemination in PDAC 4-6.”
- 6.) The following statement is also without a value “Despite the prognostic and predictive value of GATA6 loss in response to adjuvant chemotherapy 6”, authors should clearly state if this is positive or negative.
- 7.) The following statement is also without a value “Histone deacetylase inhibitors (HDACi) are known to affect EMT-like features of cancer cells and tumor-stromal crosstalk in PDAC 18,19”, authors should clearly state if this is positive or negative.

Experimental issues:

- 1.) Baseline proliferation rates should also be shown for each cell line and each treatment.
- 2.) It is unclear if the MHC-I expression correlates with active proliferation in the cells. As both GATA6 loss and MAPK activity are likely to associate with altered proliferation. The low MHC-I expression could be a result of cells in an active cell cycle, which would likely not have MHC-I on the surface as they have freshly divided. Many of the results are correlative and may not indicate a direct regulation of MHC-I.
- 3.) Figure 3: Experiments in figure 3f should be performed in the presence of anti-PD1 as a control to determine if this will eliminate the difference in between GP82 and GP99 experiments.
- 4.) Figure 5: Figure 5a, body weights (stated in results section) are missing from the figure. Figure 5b,c it is unclear why the mice died if the tumors were decreasing in volume. This should be explored and explained. Figure 5f, the MHC-I IHC is very faint if any protein is detectable at all from the image provided and is not very convincing or representative of the quantitative data in 5g. The cellular localization of GATA6 is not well explained and appears to shift from cytoplasmic to nuclear based on the treatment provided. Further fractionation analysis should be provided to quantify.
- 5.) Rationale for using the HDACi is limited and not rigorously justified. With the broad spectrum impact of HDAC inhibitors it is unclear how this study will benefit the field. The further lack of consideration for off target effects of the HDAC inhibitors is also concerning; these could have indirect effects that obscure the proposed mechanism.
- 6.) Authors fail to describe or differentiate how HDAC and MEK inhibitors directly affect T cells or other stromal cells and the implications that could have for these studies.
- 7.) Lack of validation/controls for GATA6 or MEK regulation in response to HDAC inhibitors. Authors should monitor these in response to HDAC inhibition.
nuclear versus cytoplasmic ratio of GATA6 in response to each treatment.

Reviewer #2

(Remarks to the Author)

Yang and co-authors present new data on the role of GATA6 in regulating immune responsiveness and MHC-I expression in the mouse and human models of PDAC. The study is well-designed and presents compelling evidence derived from genetically engineered cell lines and animal models. The main thrust of the paper is to demonstrate that MEKi increase MHC-I expression and type-I IFN signatures selectively in GATA6 classical subsets of PDAC, whereas GATA6-low EMT PDAC cells evade immune surveillance.

Several issues would be important to address to solidify the overall impact:

- 1) The proposed immunological mechanism of MEKi+HDACi is not sufficiently validated. Experiments to reverse the survival advantage by depletion of CD8/CD4 cells or in immunodeficient hosts will be important to exclude non-immune mechanisms of MEKi+HDACi combinatorial activity.
- 2) The ex vivo experiments with T cell and PDAC co-cultures should be feasible to model in vivo in the TCR Tg mice. This or other immunogenic models (e.g., Ova/OT-I TCR Tg) could be used to demonstrate the efficacy of MEKi+HDACi combination.
- 3) By the same token, the cellular model of GATA6 downregulation (or alternatively using Dox-shGATA6) should be deployed in vivo to validate the effect on tumor growth and MHC-I.

Minor comments:

- 1) please describe how Gata6 hi and lo cell were separated by FACS (Fig.2a).
- 2) Figure 3c and 3f are poorly visible. Suggest higher magnification. Controls of MEKi alone should be provided to demonstrate its direct effect on viability. Figure legend does not explain the color in the image- is it CFSE in T cells? Are PI images available or was it done by FACS?
- 3) Figures 6 and 7 provide indirect correlations without direct mechanistic evidence. These can be enhanced and replaced with direct results demonstrating the role of T cell immunity in the context of genetic or pharmacological GATA6 modulation.

Reviewer #3

(Remarks to the Author)

Yang et al. present evidence that GATA6+ expression in PDAC tumors and cell lines is required for MEK inhibition to promote tumor immunity via increased expression of MHC-I and IFN-gamma. However, long-term MEK inhibition in vivo is ineffective due to lower GATA6 expression and consequently, less antigen presentation. The authors show that HDAC inhibition can prevent GATA6 downregulation in tumors, and thus work synergistically with MEK inhibition. GATA6 expression may serve as a biomarker of responsiveness to MEK+HDAC inhibitor combinations. These findings are

intriguing and may have clinical relevance; however, several of the principal findings would be more convincing if supported by more in-depth characterization and by using additional cell lines.

Larger cohort and alternate analyses needed for data from patient tumors

The authors show that GATA6-high patient tumors harbor denser populations of immune cells (both anti-tumor CD8+ T cells and pro-tumor FOXP3+ T cells) compared to GATA6-low tumors. In PDAC, there is a negative correlation between tumor cell density and the densities of many immune cell types (both anti-tumor and pro-tumor). The authors should report whether tumor cellularity per se aligns with leukocyte density. Is there a relationship between tumor cellularity within groups (GATA6-high, GATA6-int, and GATA6-low)?

The GATA6 staining from CODEX in Figure 1D and E appears regional, and may be due to low-quality tissue in that region. Although the PanKRT stain is high throughout, it's difficult to tell how thresholds were set to detect staining differences. A counter stain for basal-like tumor cells in Supp Figure 1 appears to show no KRT5 staining, consistent with regional tissue quality problems. If KRT5 is not expressed by this tumor, KRT6A, and TP63 are other options. Additionally, a single patient specimen is insufficient for drawing conclusions about intratumoral GATA6 heterogeneity and leukocyte proximity. This data should be supported by simple immunofluorescence co-staining for GATA6, a basal-like marker, and 1 or 2 leukocyte markers in multiple ROIs from ≥ 5 patient tumors. Tumors with relatively uniform classical-subtype tumor cells and tumors with predominantly basal-like tumor cells should be added as controls.

The authors suggest that leukocyte density is related to GATA6 expression and not CLDN18 expression (also considered a classical subtype marker), but the comparison between CLDN18+ and CLDN18neg tumor cells in Supp Figure 1 and their proximity to immune cells is less informative than a comparison between CLDN18+GATA6neg and CLDN18negGATA6+ tumor cells. Statistics are also missing for this data.

The relationship between GATA6 expression and the classical subtype is important context, but is not sufficiently developed, especially with the patient tumor analyses. One of the many published subtyping approaches should be applied to patient tumor gene expression data, as well as to individual cells from scRNAseq datasets and LCM tumor regions. Additionally, there are several published scRNA-seq datasets that should be used to identify tumor cell-specific GATA6 expression (like the dataset used for Figure 1C). The Mauer et al. LCMD data is less ideal than scRNAseq datasets for addressing cell-level heterogeneity.

More mouse cell lines needed for in vitro mechanism experiments

The authors generated cell lines from syngeneic KPC mice. Since these lines were not subcloned, it is surprising that 3 of them show heterogeneous GATA6 expression, and 3 show no GATA6 expression at all (Figure 2A). The authors should genetically validate that these lines are exclusively tumor cells, and these experiments should be done with FACS-sorted GATA6-high and GATA6-low cells from several heterogeneous cell lines. The top histogram in Figure 2A shows an unusual bi-modal expression of GATA6 with many cells on the right y-axis. These data should be inspected for FACS artifacts, and a better example plot should be presented. Additionally, it appears that only 4 of the 6 cell lines are used for the data in Figure 2c (it's unclear how many were used for Figure 2b). How were these selected? The AID approach in Figure 2e-g more convincingly demonstrates the relationship between MEKi, GATA6, and antigen presentation, but should be performed on ≥ 3 distinct AID-altered lines. Additionally, more than 2 cell lines should be derived from the FKPC2GP mice to make conclusions that the distinct morphologies from the 2 cell lines match mechanisms related to GATA6 and MEK inhibition. Similarly, the data in Figure 4e appear to be from 2 KPC cell lines, this in vitro experiment should be performed on ≥ 5 cell lines with statistical comparisons shown.

The treatment regimen for the in vivo experiment in Supp Figure 3b is not clear. Did the recurrence occur during MEKi treatment or did MEKi withdrawal prompt the recurrence? Additionally, the authors should provide MR images from the whole cohort data.

The Figure 5e figure legend states that the MHC-I and GATA6 staining is from tumor cells but also from "whole tumorous tissue". This should be clarified. Figure 5g shows the percentage of tumor cells expressing GATA6 and MHC-I (though the y-axis is labelled only GATA6+) but it's unclear if the bars are superimposed or stacked and statistical comparisons are missing. Figure 6 shows that with the treatment effect (greatest in the MEKi+HDACi combination) there is an increase in the percentage of leukocytes. Is this the percentage among all cells? As with earlier figures, these data would be easier to interpret with cell densities.

Additional Comments

Figure 1 and throughout, text on figure labels (especially axes labels) should be written in common syntax, without underscores and periods, and with appropriate superscripts.

In Figure 1 (and throughout) for GSEA results, the authors should indicate which pathways are considered significant – the color and size scales for NES and FDR are difficult to interpret; nominal P values should also be given.

Numbers are missing from the y-axes of Figure 4b and Figure 4c making it difficult to compare data between the charts. The values for GATA6 expression are similar between the control conditions of GATA6-hi and GATA6-low tumors for Figure 4c. Did GATA6 expression change in vivo or are the y-axes different?

Reviewer #4

(Remarks to the Author)

Yang et al. investigated the impact of combined MEK inhibitor (MEKi) and histone deacetylase inhibitor (HDACi) treatment on GATA6-dependent MHC1 expression in pancreatic ductal adenocarcinoma (PDAC). The authors showed that MEKi alone increased MHC1 expression in GATA6-positive pancreatic cancer cells and triggers epithelial-to-mesenchymal transition that appeared to reduce the effect. The addition of HDACi (domatinostat), preserved the GATA6-positive population and enhanced MHC1 expression, resulting in improved immune response and decreased tumor growth. This combination therapy seemed to significantly increase the cytotoxic T-cell infiltration, enhancing anti-tumor responses. However, there appears to be a lack of coherence and consistency in linking the mechanistic insights across clinical data and experimental data. More importantly, it is not clear that MHC1 up-regulation is mechanistically important for the anti-tumor immunity upon the MEKi/HDACi combo treatment, while MEKi/HDACi combo treatment might have pleiotropic effects on both tumor and stroma. While the study presents a novel approach to enhancing immunogenicity in PDAC, the evidence supporting the main hypotheses needs to be significantly improved. The detailed points are listed below.

Major points

1. The clinical data analysis in Figure 1 shows an association between GATA6 expression and the inflammatory response signature including MHC1 up-regulation. However, it remains unclear how these findings relate to the therapeutic context of MEKi, since the patient dataset indicates these associations exist independent of MEK inhibition. The observation on clinical data is consistent with preclinical models? GATA6 positive cell lines tend to have the inflammatory response signature and MHC1 up-regulation (without MEKi)? As another example, in GATA6high PDX vs. GATA6low PDX in figure 4, do they have different MHC1 expression level?
2. The induction of MHC1 in GATA6+ pancreatic cancer cells upon MEKi treatment was shown in a couple of KPC cell lines. However, this does not appear robust enough or whether similar dynamics occur in human PDAC cell lines across different molecular subtypes (PDAC cell lines known to belong to classical subtype vs. basal-like).
3. In figure 3, the analysis seems limited as it tests GATA6-dependent MHC1 expression and T-cell cytotoxicity mainly in one cell line each. Expanding this to more diverse cell lines could strengthen the conclusions drawn. It is not clear whether differential expression of MHC1 is functionally important to determine anti-tumor activity.
4. The introduction of HDAC inhibitors is abrupt and their role in the combinatory treatment effect alongside MEKi is not sufficiently explained.
5. When combo-treatment is done in PDX models (Figure 4), is the therapeutic effect mainly through MHC1-T cell mediated cytotoxicity? If so, PDX models in immune-compromised or -deficient hosts show no obvious difference since they lack adaptive immune system?
6. Figure 5e data seems contradictory with what the authors showed so far. MEKi treatment did not up-regulate MHC1 expression. HDACi treatment didn't seem to retain GATA6 positive population either. Only combo context resulted in up-regulation of MHC1 (no change in GATA6 positive population). This also raise a concern whether the main hypothesis was thoroughly tested and the conclusion is supported by robust data.

Minor points

1. In Figure 3a, the figure is not complete.
2. In figure 4, the graphs do not have some information such as y-axis titles and units.
3. In Figure 4, for the PDX experiments, the authors used mocetinostat as HDACi, and then switched to dometinostat. Any reason?
4. Figure 5e & 5g, it is questionable whether it accurately counted GATA6+ and MHC1+ cells. The positive cell population ranges only 2-8% in the entire population in a given area?

Version 1:

Reviewer comments:

Reviewer #1

(Remarks to the Author)

The authors have sufficiently addressed my original concerns.

One additional concern is over Reference 35, which is bioRxiv from 2022. It is unclear whether Nature allows non-peer review references. Additionally, the age of this reference is also concerning, because it has not been officially published for over 3 years. I would recommend that essential data be added as supplemental for this manuscript or anything associated with that reference be removed as it has not been adequately peer reviewed.

Reviewer #2

(Remarks to the Author)

The authors exhaustively addressed all critical comments from my and other reviewers. I believe the manuscripts is

substantially improved and can be an exciting contribution to the field.

Reviewer #3

(Remarks to the Author)

The authors have done a nice job addressing the concerns from the initial review. They've clarified the writing, added important new experiments (including the GATA6 knockout/AID models and CD8 depletion studies), and expanded both the cell line and patient tumor data to strengthen the conclusions. The rationale for including HDAC inhibition is now much clearer, and they've been transparent about its limitations.

Reviewer #4

(Remarks to the Author)

In the revised manuscript, the authors have provided more robust data to support GATA6-dependent MHC1 expression in the context of MEK inhibition. The new data clearly strengthen the conclusion of the revised manuscript. Although the authors effectively showed that GATA6 expression is associated with a more immunogenic basal state, GATA6 does not seem to be responsible for up-regulation of MHC1 expression in the basal state as shown in Figure 2d-h. GATA6 knock-outs or degradation have no effect on the basal level of MHC1, while GATA6-high vs. low cells showed a trend of MHC1 expression difference. This difference should be clearly documented and acknowledged. The authors have addressed all other concerns in the revised manuscript.

REVIEWER COMMENTS

Reviewer #1 (Remarks to the Author): with expertise in pancreatic cancer

The authors present a manuscript that assesses the ability of MEK inhibitors to alter MHC1 levels depending upon the level of GATA6 in pancreatic cancer cell lines. The present a convincing MFI analysis of two high GAT6 versus two low GAT6 cell lines to demonstrate only the high GATA6 cell lines increase MHC1 when exposed to MEKi. As MHC1 is critical for immune recognition and/or evasion, mechanisms that can increase MHC1 expression are critical for improving immunotherapy. The authors further combine MEKi with HDACi, which shows increased survival of CKP pancreatic cancer bearing mice. The mechanisms behind the HDACi is not well laid out or tested, leaving a GAP in the understanding and applicability of this combination therapy. This is a highly correlative study that provides novel insight into the roles of MEK and GATA6 in pancreatic cancer, yet multiple moderate concerns decrease the enthusiasm for this manuscript.

Response:

→ We thank the reviewer for the thorough review and careful, well-stated concerns and suggestions. As described below, we have significantly extended our data and modified the manuscript to add depth and give more details.

Concerns:

Writing issues:

1.) The introduction and some of the discussion are not well written and lack clarity for multiple statements; they are generally broad and do not provide the reader with accurate/specific interpretations from the cited references. Especially with respect to previous studies on GATA6 in the introduction. Additionally, the First two lines in the abstract are too broad, unsupported, and should be rewritten.

Response:

→ Thank you for your comments. We have substantially revised the Introduction, Abstract, and relevant parts of the Discussion to improve clarity, specificity, and scientific accuracy. In particular, the first two lines of the abstract have been rewritten to avoid overly broad statements and better reflect the content and significance of the study.

2.) Multiple small grammatical issues are detected throughout the manuscript.

Response:

→ We have checked for grammatical mistakes thoroughly and revised accordingly in the revised version.

3.) Methods for ultrasound imaging of the tumors is not provided.

Response:

→ The methods for ultrasound imaging have been previously described in detail in our published study ¹. Additionally, we have now included representative ultrasound images in Fig. 3c of the revised manuscript.

4.) GP99 cells should not be portrayed as GATA6⁻ as this implies negative; in reality they do express low levels (fig3d). It is suggested to use the term GATA6^{lo}. It is also suggested that a GATA6 knockout be test as cancer cells are highly plastic and the ability to reexpress GATA6 in lo cells is unclear.

Response:

→ We thank the reviewer for the helpful suggestions. In the revised manuscript, the original Figure 3 has been reorganized and is now presented as Figure 5. In the revised manuscript, we have updated our terminology to describe GP99 and similar cell lines as GATA6^{low} rather than GATA6⁻, to more accurately reflect their low but detectable GATA6 expression, as shown in Fig. 5c (original Fig. 3d).

To directly address the reviewer's concern regarding cancer cell plasticity and the potential for GATA6 re-expression, we generated GATA6 knockout lines using CRISPR-Cas9 in the GATA6^{high} cell line 2838c3 (Fig. 2d–e, Fig. S2b). We derived two independent knockout clones (gGATA6_KO-3 and gGATA6_KO-4), and found that the MEKi-induced upregulation of MHC1 seen in the parental line was largely abolished in the KO clones. These results demonstrate that GATA6 is required for the MEKi-mediated induction of MHC1 *in vitro*.

To validate the role of GATA6 *in vivo*, we employed a targeted GATA6 protein degradation system using the orthotopic transplantation model of auxin-inducible degron GATA6 knock-in (AID-GATA6-KI) cells derived from 110299, another GATA6^{high} cell line, rather than GATA6-KO cells (Fig. 3e-h, S3d-f). This approach was chosen because GATA6's strong interaction with the immune microenvironment could potentially alter tumor formation and baseline tumor microenvironment (TME) in GATA6-KO models, complicating the assessment of drug-induced effects and leading to different conclusions. With the AID-GATA6-KI system, we were able to reduce GATA6 protein expression after tumor establishment, and GATA6 levels remained low throughout the treatment period upon auxin administration (Fig. 3g, S3d). We found that GATA6 degradation significantly impaired MEKi-induced tumor suppression in the 110299 orthotopic model (Fig. 3f). Additionally, the MEKi-induced upregulation of tumor MHC1 expression, CD8 infiltration, and anti-tumor cytotoxicity were all markedly reduced (Fig. 3h), highlighting GATA6's essential role in MEKi-induced anti-tumor responses. Notably, MEKi-induced CD4 infiltration was unaffected by GATA6 degradation (Fig. S3f), suggesting that GATA6 specifically mediates the MEKi effect via the MHC1-CD8 pathway.

5.) GATA6 does not “orchestrate” EMT, instead the reference demonstrates that GATA6 is required to maintain epithelial differentiation. The following is therefore quite misleading and should be rewritten : “Recent seminal studies have illuminated the crucial tumor-suppressive role of GATA6 in orchestrating epithelial-to-mesenchymal transition (EMT) and restraining tumor dissemination in PDAC 4-6.”

Response:

→ We have revised the text as “GATA6 is a key regulator of the classical PDAC phenotype, maintaining epithelial differentiation while restraining epithelial-to-mesenchymal transition (EMT)” in page 6 line 2-3.

6.) The following statement is also without a value “Despite the prognostic and predictive value of GATA6 loss in response to adjuvant chemotherapy 6”,

authors should clearly state if this is positive or negative.

Response:

→ We have revised the text to provide a more specific statement “Retrospective studies have shown strong associations between high GATA6 expression and prolonged survival in treatment-naive and advanced PDAC cases” in page 6 line 5-6.

7.) The following statement is also without a value “Histone deacetylase inhibitors (HDACi) are known to affect EMT-like features of cancer cells and tumor-stromal crosstalk in PDAC 18,19”, authors should clearly state if this is positive or negative.

Response:

→ We have revised the text to provide a more confirmative statement as “Histone deacetylase inhibitors (HDACi) are known to promote epithelial gene expression in tumor cells and reduce pro-tumorigenic stromal activation in PDAC” in page 7 line 3-4.

Experimental issues:

1.) Baseline proliferation rates should also be shown for each cell line and each treatment.

Response:

→ We thank the reviewer for this important suggestion. In the revised manuscript, we performed a BrdU assay to assess the proliferation status of all cell lines *in vitro* (Fig. S2a). While we observed differences in baseline proliferation rates among the cell lines, no correlation was found with their baseline GATA6 expression levels. For MEKi-treated cells, there was a noticeable but statistically non-significant reduction in proliferation (Fig. S2a), and the extent of this decrease was not associated with GATA6 expression changes.

2.) It is unclear if the MHCI expression correlates with active proliferation in the cells. As both GATA6 loss and MAPK activity are likely to associate with altered proliferation. The low MHCI expression could be a result cells in an active cell cycle, which would likely not have MHCI on the surface as they have freshly divided. Many of the results are correlative and may not indicate a direct regulation of MHCI.

Response:

→ As mentioned above, we assessed the proliferation status of the cells both at baseline and after treatment. No significant correlation was observed between proliferation status and GATA6 expression, or between proliferation status and MHCI levels (Fig. S2a, Fig. 2a,2c). Additionally, we examined the co-expression of MHCI and GATA6 with the proliferative marker PCNA in our human PDAC specimens (n=8) and found no correlation between proliferation status and MHCI and/or GATA6 expression (Fig. S1d).

3.) Figure 3: Experiments in figure 3f should be performed in the presence of anti-PD1 as a control to determine if this will eliminate the difference in between GP82 and GP99 experiments.

Response:

→ We thank the reviewer for this thoughtful comment. In the revised manuscript, the original Figure 3 has been reorganized and is now presented as Figure 5.

In these short-term co-culture experiments (2–3 days), we use PD1 primarily as a T cell activation marker rather than a marker of exhaustion, as exhaustion typically arises in chronic stimulation contexts. Therefore, we do not expect functional exhaustion to be a confounding factor in these assays.

Although we observed a significant increase in PD1⁺ cells following combination treatment *in vitro* (Fig. 5e,f) and *in vivo* (Fig. 7a), we did not include anti-PD1 therapy in these experiments. This was because cytotoxic activity remained robust, as evidenced by high GzmB⁺ cell frequencies—and tumor growth was already well controlled, indicating that additional PD1 blockade might not further enhance anti-tumor effects.

In fact, the limiting factor for the combination treatment *in vivo* was not tumor progression but treatment-related toxicity, as evidenced by significant weight loss (Fig. S7a), which led to early termination of the experiment despite minimal tumor burden. Importantly, the central aim of our study is to define the immunological role of GATA6 expression in shaping therapeutic responses, particularly through its influence on tumor cell state plasticity. The MEKi/HDACi combination serves as a proof-of-concept strategy to preserve the GATA6⁺ tumor population and enhance immune responsiveness, thereby supporting our broader hypothesis that GATA6-mediated tumor cell states are key determinants of treatment efficacy.

4.) Figure 5: Figure 5a, body weights (stated in results section) are missing from the figure. Figure 5b,c it is unclear why the mice died if the tumors were decreasing in volume. This should be explored and explained. Figure 5f, the MHC IHC is very faint if any protein is detectable at all from the image provided and is not very convincing or representative of the quantitative data in 5g. The cellular localization of GATA6 is not well explained and appears to shift from cytoplasmic to nuclear based on the treatment provided. Further fractionation analysis should be provided to quantify.

Response:

→ We appreciate the reviewer's detailed observations. The original Figure 5 has now been reorganized and is presented as Figure 6 in the revised manuscript.

→ Regarding the body weight data, we have now included the data in the revised supplementary figure (Fig. S7a). In the *in vivo* experiment with combination treatment, mice were sacrificed due to significant weight loss, despite small tumor burdens. As the data indicate, MEKi alone didn't lead to significant body weight loss, whereas the combination of MEKi and HDACi induced substantial weight loss. These findings are consistent with previous reports of HDACi-associated adverse effects, such as gastrointestinal toxicity and constitutional symptoms, which have been observed in clinical trials². While HDACi toxicity limited treatment tolerability, its use in our study was primarily as a mechanistic tool to restore and maintain the GATA6⁺ tumor cell population, thereby enabling us to test the hypothesis that preserving this cell state can enhance therapeutic responses.

→ Regarding MHC I staining, to better visualize the increase in MHC I expression on GATA6⁺ tumor cells upon MEKi and HDACi treatment, we performed mIF on all four treatment groups, allowing comparison of MHC I expression and its co-expression with GATA6. High-resolution mIF images are included in the revised manuscript (Fig. 6f), and the quantification corresponds well with the observed staining patterns.

→ Regarding the GATA6 localization, since GATA6 is a transcription factor, its signal should be localized mainly to the nucleus. To address cytoplasmic staining observed in the original manuscript (Fig. 5e), we repeated the staining using an optimized

protocol in the revised manuscript (Fig. 6e), which yielded nuclear-specific GATA6 signals. All quantifications were based exclusively on nuclear GATA6 staining. Given that the cytoplasmic signal was artifactual and has now been corrected with specific nuclear staining, we believe additional subcellular fractionation is not required at this point.

5.) Rationale for using the HDACi is limited and not rigorously justified. With the broad spectrum impact of HDAC inhibitors it is unclear how this study will benefit the field. The further lack of consideration for off target effects of the HDAC inhibitors is also concerning; these could have indirect effects that obscure the proposed mechanism.

Response:

→ We thank the reviewer for this important comment. We agree that HDAC inhibitors have broad epigenetic effects and that mechanistic clarity is critical when interpreting their impact. In our study, HDAC inhibition was employed as a proof-of-concept tool to restore the GATA6⁺ tumor cell population that is responsive to MEKi treatment. This approach demonstrates that modulating tumor cell states can enhance therapeutic outcomes.

We selected HDACi based on prior studies showing that HDAC activity promotes EMT, and that HDAC inhibition can reinforce epithelial lineage programs in PDAC, including GATA6 expression (e.g., through BRD4- and MYC-dependent mechanisms as described for Domatinostat)³. Since MEKi treatment consistently reduced GATA6 expression and promoted EMT features, we hypothesized that HDAC inhibition could stabilize GATA6 levels and thereby preserve a MEKi-responsive tumor cell population. Indeed, we demonstrated that HDACi treatment largely restores GATA6 expression *in vitro* and *in vivo*, significantly improving tumor control and mouse survival. The revised manuscript includes data confirming GATA6 restoration by HDACi both *in vitro* (Fig 4d-e) and *in vivo* (Fig. 6e-f, S5d).

To address potential off-target or indirect immune effects of HDACi, we compared the immune-related transcriptomic profiles of *CKP* tumors treated with or without MEKi and HDACi using the NanoString PanCancer Immune Profiling Panel. Gene set variation analysis (GSVA) showed that MEKi monotherapy induced several immune-related gene sets including antigen receptor signaling, T cell activation, cytotoxicity, and type II interferon signaling (Fig. 6d). In contrast, HDACi alone had a modest impact, enriching for antigen processing and IFN- γ -related pathways. Notably, the combination therapy activated a broader range of immune pathways than either monotherapy, including additive or synergistic enrichment of the pathways noted above (Fig. 6d).

Further differential expression analysis between MEKi and combination-treated tumors revealed that the combination therapy significantly upregulated genes associated with T cell function and antigen presentation (Fig. S7b,c). Additionally, the combination group exhibited higher scores in pathways related to antigenicity, such as MHC and antigen processing (Fig. S7b, c), suggesting that combined MEKi and HDACi enhances tumor antigenicity compared to MEKi alone. Taken together, these data support our use of HDACi as a mechanistic tool to modulate tumor cell state, restore GATA6 expression, and improve immunogenic responses to MEKi. We have clarified this rationale and expanded the description of the transcriptomic data in the Results section of the revised manuscript.

In addition, the toxicity profile of HDAC inhibitors is a critical limitation for their broader clinical use and is discussed in more detail in the revised Discussion section. Consistent with clinical observations, in our *in vivo* experiments the combination of

MEKi and HDACi induced significant weight loss, resulting in early termination of the study despite small tumor burdens (Fig. S7a). MEKi treatment alone did not cause significant body weight loss. Similar gastrointestinal and systemic toxicities leading to weight loss have been reported in patients treated with HDAC inhibitors ². Despite these effects, HDACi was used in this study primarily as a tool to restore the GATA6⁺ population responsive to treatment, serving as a proof-of-principle experiment demonstrating that this strategy could enhance therapeutic responses.

6.) Authors fail to describe or differentiate how HDAC and MEK inhibitors directly affect T cells or other stromal cells and the implications that could have for these studies.

Response:

→ We agree with the reviewer and have worked intensively on better characterizing the role of both targeting approaches. In order to delineate the effects induced by HDACi and MEKi separately, we analyzed the immune transcriptomic profiles of *CKP* tumors treated by the single treatment of the two inhibitors (Fig. 6d). Gene set variation analysis revealed that MEKi alone induced the expression of several immune-related gene sets, including antigen receptor-mediated signaling, T cell receptor signaling, type II interferon activation, T cell activation, and cytotoxicity (Fig. 6d), suggesting its role in promoting T cell-mediated immunity. In contrast, HDACi alone had a more subtle effect on the tumor immune microenvironment, inducing pathways related to antigen processing and type II interferon regulation (Fig. 6d).

We have included the above data in the revised manuscript (Fig. 6d).

7.) Lack of validation/controls for GATA6 or MEK regulation in response to HDAC inhibitors. Authors should monitor these in response to HDAC inhibition. nuclear versus cytoplasmic ratio of GATA6 in response to each treatment.

Response:

→ We thank the reviewer for highlighting the need to validate GATA6 changes upon HDAC inhibition. To address this, we evaluated GATA6 expression changes in response to HDAC inhibitors both *in vitro* (Fig. 4d-e) and *in vivo* (Fig. 6e-f). Our data show that HDAC inhibition restores GATA6 expression without causing significant fluctuations in overall levels.

Regarding GATA6 expression, since GATA6 is a transcription factor, its signal should be predominantly nuclear. In the original manuscript (Fig. 5e), some cytoplasmic staining was observed; to clarify this, we optimized the immunostaining protocol for the revised manuscript, which now yields specific nuclear GATA6 signals (Fig. 6e). All quantifications were performed exclusively on nuclear GATA6 staining. Since the previous cytoplasmic signal was artifactual and corrected by the improved protocol, we believe that additional nuclear vs cytoplasmic quantification is not necessary.

Reviewer #2 (Remarks to the Author): with expertise in pancreatic cancer

Yang and co-authors present new data on the role of GATA6 in regulating immune responsiveness and MHC-I expression in the mouse and human models of PDAC. The study is well-designed and presents compelling evidence derived from genetically engineered cell lines and animal models. The main thrust of the paper is to demonstrate that MEKi increase MHC-I expression and type-I IFN

signatures selectively in GATA6 classical subsets of PDAC, whereas GATA6-low EMT PDAC cells evade immune surveillance.

Response:

→ We sincerely thank the reviewer for the positive evaluation of our study and for recognizing the significance of our findings.

Several issues would be important to address to solidify the overall impact:

1) The proposed immunological mechanism of MEKi+HDACi is not sufficiently validated. Experiments to reverse the survival advantage by depletion of CD8/CD4 cells or in immunodeficient hosts will be important to exclude non-immune mechanisms of MEKi+HDACi combinatorial activity.

Response:

→ We thank the reviewer for this important suggestion and agree that further validation of the immunological mechanisms underlying MEKi and HDACi combination therapy is essential. Accordingly, we have expanded our experimental efforts to clarify the immune contributions in greater detail.

First, we utilized the AID-GATA6-KI system to inducibly reduce GATA6 protein expression after tumor establishment. Upon auxin administration, GATA6 levels remained low throughout treatment (Fig. 3g, S3d). We observed that GATA6 degradation significantly impaired MEKi-induced tumor suppression in the 110299^{AID-GATA6} orthotopic model (Fig. 3f). Furthermore, the MEKi-driven upregulation of tumor MHC1, CD8⁺ T cell infiltration, and anti-tumor cytotoxicity were markedly diminished (Fig. 3h), highlighting GATA6's essential role in MEKi-induced anti-tumor responses. Notably, MEKi-induced CD4 infiltration was unaffected by GATA6 degradation (Fig. S3f), suggesting that GATA6 specifically mediates the MEKi effect via the MHC1-CD8 pathway.

Second, we evaluated the combined MEKi and HDACi treatment in both the 110299^{AID-GATA6} orthotopic model (Fig. S6a,c,d) and *CKP* tumor-bearing mice (Fig. 6). Compared to MEKi monotherapy, which reduced GATA6 expression but increased MHC1 levels, the combination treatment restored GATA6 expression while maintaining high levels of MHC1 (Fig. 6e-f, S6a,d) and resulted in stronger tumor control (Fig. 6b-c, S6c).

In order to confirm the role of CD8, we performed additional experiment to deplete CD8 cells with anti-CD8 antibody (Fig. S6b-d), and here we observed that CD8 depletion greatly abolished the tumor-suppressive effect of the combined treatment. Histological analysis revealed that tumor cell GATA6 expression remained largely unchanged, while MHC1 expression stayed elevated following MEKi and HDACi combination therapy, regardless of CD8 status, suggesting that MHC1 upregulation is independent of CD8⁺ T cell activity (Fig. S6d). In contrast, the increase of Cl casp3 after MEKi and HDACi combination therapy was abolished when CD8⁺ T cells were depleted, further supporting the critical role of CD8⁺ T cells in mediating the anti-tumor effects of the combined MEK and HDAC inhibition.

These new data have been incorporated into the revised manuscript (Fig.3; Fig.S3e-f; Fig. 6b-f; Fig. S6).

2) The ex vivo experiments with T cell and PDAC co-cultures should be feasible to model in vivo in the TCR Tg mice. This or other immunogenic models (e.g., Ova/OT-I TCR Tg) could be used to demonstrate the efficacy of MEKi+HDACi combination.

Response:

→ To demonstrate the efficacy of MEKi and HDACi combination *in vitro*, we have included the combination treatment in the co-culture experiments of the gp33-expressing tumor cells with the gp33-TCR transgenic T cells. Besides, in the revised manuscript, we included 2 additional gp-expressing murine PDAC cell lines. The results showed that combined MEKi and HDACi treatment induces MHC-I upregulation on PDAC cells, which was translated into enhanced tumor-specific T cell cytotoxicity. We have included the results in the revised manuscript in Fig. 5.

3) By the same token, the cellular model of GATA6 downregulation (or alternatively using Dox-shGATA6) should be deployed *in vivo* to validate the effect on tumor growth and MHC-I.

Response:

→ To validate the role of GATA6 *in vivo*, we employed a targeted GATA6 protein degradation system using the orthotopic transplantation model of auxin-inducible degron GATA6 knock-in (AID-GATA6-KI) cells derived from 110299, another GATA6^{high} cell line, instead of complete GATA6 knockout cells (Fig. 3e-h, S3d-f). We chose this approach because the strong interaction of GATA6 with the immune microenvironment could profoundly alter tumor initiation and baseline tumor microenvironment in GATA6 knockout models, complicating the interpretation of drug-induced effects and potentially leading to confounding conclusions.

With the AID-GATA6-KI system, we achieved inducible reduction of GATA6 protein expression after tumor establishment, with sustained low GATA6 levels throughout the treatment period upon auxin administration (Fig. 3g, S3d). We observed that GATA6 degradation significantly impaired MEKi-induced tumor suppression in the 110299 orthotopic model (Fig. 3f). Additionally, the MEKi-induced upregulation of tumor MHC-I expression, CD8 infiltration, and anti-tumor cytotoxicity were all markedly reduced (Fig. 3h), highlighting GATA6's essential role in MEKi-induced anti-tumor responses. Notably, MEKi-induced CD4 infiltration was unaffected by GATA6 degradation (Fig. S3f), indicating that GATA6 specifically influences the MEKi effect via the MHC-I-CD8⁺ T cell axis.

Minor comments:

1) please describe how Gata6 hi and lo cell were separated by FACS (Fig.2a).

Response:

→ Cells were fixed, permeabilized, and stained with a GATA6-specific antibody or the corresponding isotype control. Flow cytometric analysis was then performed to measure GATA6 levels. Based on the median fluorescence intensity (MFI) of GATA6 staining, cells were classified into GATA6^{high} and GATA6^{low} populations, as shown in Fig. 2a.

2) Figure 3c and 3f are poorly visible. Suggest higher magnification. Controls of MEKi alone should be provided to demonstrate its direct effect on viability. Figure legend does not explain the color in the image- is it CFSE in T cells? Are PI images available or was it done by FACS?

Response:

→ In the revised manuscript, the original Figure 3 has been reorganized and is now presented as Figure 5. We have replaced the relevant images with higher-resolution versions to improve visibility (Fig. 5e, f), as suggested.

To address the reviewer's concern regarding MEKi controls, we have now included cytotoxicity data from gp-expressing tumor cells treated with MEKi and/or HDACi in the presence (Fig. 5e, f) or absence of gp-TCR T cells (Fig. S5h). These results demonstrate that in the absence of T cells, the treatment does not induce statistically significant cytotoxicity, indicating that the observed effects are largely T cell-mediated rather than a direct consequence of MEKi or HDACi treatment alone.

→ We have also revised the figure legend to clarify the color coding: the green signal corresponds to CFSE-labeled T cells.

→ Regarding PI staining, detection of PI⁺ tumor cells was performed by flow cytometric analysis, and this information has been added to the corresponding figure legend (page 61, line 19-20) for clarity.

3) Figures 6 and 7 provide indirect correlations without direct mechanistic evidence. These can be enhanced and replaced with direct results demonstrating the role of T cell immunity in the context of genetic or pharmacological GATA6 modulation.

Response:

→ To provide more direct mechanistic evidence of the role of T cell immunity in the context of GATA6 modulation, we expanded our experimental approaches in the revised manuscript.

First, we included the AID-GATA6-KI system, which was able to reduce GATA6 protein expression after tumor establishment, and GATA6 levels remained low throughout the treatment period upon auxin administration (Fig. 3g, S3d). We found that GATA6 degradation significantly impaired MEKi-induced tumor suppression in the 110299^{AID-GATA6} orthotopic model (Fig. 3f). Additionally, the MEKi-induced upregulation of tumor MHC-I expression, CD8 infiltration, and anti-tumor cytotoxicity were all markedly reduced (Fig. 3h), highlighting GATA6's essential role in MEKi-induced anti-tumor responses. Notably, MEKi-induced CD4 infiltration was unaffected by GATA6 degradation (Fig. S3f), suggesting that GATA6 specifically mediates the MEKi effect via the MHC-I-CD8 pathway.

Second, we evaluated the combined MEKi and HDACi treatment in both the 110299 orthotopic model (Fig. S6a,c,d) and *CKP* tumor-bearing mice (Fig. 6). Compared to MEKi monotherapy (Fig. S4a; Fig. 3d), which reduced GATA6 expression (Fig. S4a) but increased MHC-I levels (Fig. 3d), the combination treatment restored GATA6 expression while maintaining high levels of MHC-I (Fig. 6e-f, S6d) and resulted in stronger tumor control (Fig. 6b-c, S6c).

In order to confirm the role of CD8, we performed additional experiment to deplete CD8 cells with anti-CD8 antibody (Fig. S6b-d), and here we observed that CD8 depletion greatly abolished the tumor-suppressive effect of the combined treatment. Histological analysis revealed that tumor cell GATA6 expression remained largely unchanged, while MHC-I expression stayed elevated following MEKi and HDACi combination therapy, regardless of CD8 status, suggesting that MHC-I upregulation is independent of CD8⁺ T cell activity (Fig. S6d). In contrast, increase of Cl casp3 after MEKi and HDACi combination therapy was abolished when CD8⁺ T cells were

depleted, further supporting the critical role of CD8⁺ T cells in mediating the anti-tumor effects of the combined MEK and HDAC inhibition.

These data provide direct mechanistic evidence that GATA6 modulates anti-tumor immunity primarily via MHC-I-mediated CD8⁺ T cell responses, and have been incorporated into the revised manuscript (Fig. 3; Fig. S3e-f; Fig. 6b-f; Fig. S6).

Reviewer #3 (Remarks to the Author): with expertise in pancreatic cancer

Yang et al. present evidence that GATA6⁺ expression in PDAC tumors and cell lines is required for MEK inhibition to promote tumor immunity via increased expression of MHC-I and IFN-gamma. However, long-term MEK inhibition in vivo is ineffective due to lower GATA6 expression and consequently, less antigen presentation. The authors show that HDAC inhibition can prevent GATA6 downregulation in tumors, and thus work synergistically with MEK inhibition. GATA6 expression may serve as a biomarker of responsiveness to MEK+HDAC inhibitor combinations. These findings are intriguing and may have clinical relevance; however, several of the principal findings would be more convincing if supported by more in-depth characterization and by using additional cell lines.

Response:

→ We sincerely thank the reviewer for the thoughtful summary and constructive comments. We appreciate the recognition of the potential clinical relevance of our findings, particularly regarding the role of GATA6 in modulating tumor immune responses to MEK and HDAC inhibition. We fully agree that further in-depth characterization and validation across additional models would strengthen the conclusions, and we have taken steps to address these points in the revised manuscript.

Larger cohort and alternate analyses needed for data from patient tumors.

The authors show that GATA6-high patient tumors harbor denser populations of immune cells (both anti-tumor CD8⁺ T cells and pro-tumor FOXP3⁺ T cells) compared to GATA6-low tumors. In PDAC, there is a negative correlation between tumor cell density and the densities of many immune cell types (both anti-tumor and pro-tumor). The authors should report whether tumor cellularity per se aligns with leukocyte density. Is there a relationship between tumor cellularity within groups (GATA6-high, GATA6-int, and GATA-6-low)?

Response:

→ We thank the reviewer for this insightful point and fully agree on the need to clarify the relationship between GATA6 expression, tumor cellularity, and immune cell infiltration. To address this, we analyzed the correlation between the percentage of CK19⁺ area per total tissue core area (as for tumor cellularity), GATA6 expression levels, and the densities of various immune cell types. Our analysis revealed a weak but statistically significant positive correlation between GATA6 expression and tumor cellularity. However, none of the immune cell markers showed a negative correlation with tumor cellularity (see Fig. 1 and Table 1 for reviewers). Notably, CD20 and CD68 densities in the stroma were even positively correlated with high tumor cellularity (Fig. 1 and Table 1 for reviewers). It is important to note that for all stromal-restricted immune markers, cell densities were normalized to the stromal area, ensuring that

observed differences in immune cell abundance are not confounded by variations in tumor cellularity.

The GATA6 staining from CODEX in Figure 1D and E appears regional, and may be due to low-quality tissue in that region. Although the PanKRT stain is high throughout, it's difficult to tell how thresholds were set to detect staining differences. A counter stain for basal-like tumor cells in Supp Figure 1 appears to show no KRT5 staining, consistent with regional tissue quality problems. If KRT5 is not expressed by this tumor, KRT6A, and TP63 are other options. Additionally, a single patient specimen is insufficient for drawing conclusions about intratumoral GATA6 heterogeneity and leukocyte proximity. This data should be supported by simple immunofluorescence co-staining for GATA6, a basal-like marker, and 1 or 2 leukocyte markers in multiple ROIs from ≥ 5 patient tumors. Tumors with relatively uniform classical-subtype tumor cells and tumors with predominantly basal-like tumor cells should be added as controls.

Response:

→ We fully agree that appropriate thresholds, biological controls, and sufficient sample size are critical to support our conclusions.

To address these concerns, we have now included data from 7 additional patient tumors subjected to multiplex immunofluorescence and spatial analysis. This includes two resected tumors, each with four representative ROIs, and six additional tumors from a tissue microarray (TMA), in which full core sections were analyzed. For each case, average values from the multiple ROIs or TMA cores are shown. Clinicopathological characteristics of all patients are now provided in Supplementary Table 5. The inter-patient heterogeneity of GATA6 expression is evident in Fig. S1e, supporting the biological variability of this marker in human PDAC.

To assess immune cell distribution and proximity to tumor subtypes, we analyzed 10 markers, including GATA6, KRT5, and key immune markers such as CD4, CD8, CD20, and CD68. This allowed us to examine spatial relationships between tumor cells and leukocyte populations across multiple tumor contexts. These additional data are presented in revised Fig. 1 and Supplementary Fig. S1.

Regarding the concern about possible tissue quality artifacts in the original sample, we quantified KRT5⁺ cells across all eight tumor samples (Fig. S1e). We found that three tumors exhibited positive KRT5 expression, while the remaining did not, indicating that the absence of KRT5 in the original index case reflects tumor-intrinsic biology rather than technical failure. We appreciate the reviewer's suggestion and have added this quantification to the revised manuscript.

These new data, presented in Fig. 1 and Supplementary Fig. S1, provide a more comprehensive foundation for the conclusions drawn in the manuscript.

The authors suggest that leukocyte density is related to GATA6 expression and not CLDN18 expression (also considered a classical subtype marker), but the comparison between CLDN18⁺ and CLDN18^{neg} tumor cells in Supp Figure 1 and their proximity to immune cells is less informative than a comparison between CLDN18⁺GATA6^{neg} and CLDN18^{neg}GATA6⁺ tumor cells. Statistics are also missing for this data.

Response:

→ In the revised manuscript, we have reanalyzed the data by comparing between GATA6⁺CLDN18⁻ and GATA6⁻CLDN18⁺ tumor cell populations (Fig. S1e, f). This

refined analysis revealed that GATA6⁺CLDN18⁻ tumor cells exhibit significantly stronger spatial interactions with CD8⁺ T cells, whereas GATA6⁻CLDN18⁺ cells do not show such enrichment (Fig. S1f). A recent publication highlighted the role of tumor-derived CLDN18 in promoting T lymphocyte infiltration and antitumor immunity in PDAC ⁴.

Our analysis shows a high degree of co-expression between GATA6 and CLDN18 in PDAC tumor cells (Fig. S1e). This co-expression may have confounded previous interpretations attributing immune cell proximity solely to CLDN18. The specific enrichment of CD8⁺ T cells near GATA6⁺CLDN18⁻ cells (Fig. S1f) suggests that GATA6, rather than CLDN18, may play a dominant role in orchestrating tumor–T cell interactions. Statistical comparisons have been added to Fig. S1f to support these conclusions.

The relationship between GATA6 expression and the classical subtype is important context, but is not sufficiently developed, especially with the patient tumor analyses. One of the many published subtyping approaches should be applied to patient tumor gene expression data, as well as to individual cells from scRNAseq datasets and LCM tumor regions. Additionally, there are several published scRNA-seq datasets that should be used to identify tumor cell-specific GATA6 expression (like the dataset used for Figure 1C). The Mauer et al. LCMD data is less ideal than scRNAseq datasets for addressing cell-level heterogeneity.

Response:

→ We appreciate the reviewer's thoughtful suggestion. In this study, we focus on the epithelial features of GATA6⁺ tumor cells rather than defining GATA6 as a classical subtype marker. While many studies support GATA6 as a marker of the classical PDAC subtype, accumulating evidence suggests substantial plasticity within PDAC subtypes, including the potential for tumor cells to transition between classical- and basal-like programs. This dynamic nature complicates rigid subtype assignments and supports a more nuanced view of tumor cell states.

In line with this, our spatial analysis of human tumors (Fig. S1e) shows that GATA6 and the classical marker CLDN18 are frequently, but not universally, co-expressed, indicating heterogeneity even within putative classical tumors. Thus, rather than applying a predefined subtyping framework, we focus on GATA6⁺ tumor cells as a biologically relevant subpopulation with epithelial features and immune relevance.

We agree that scRNAseq datasets are highly informative for characterizing tumor heterogeneity. In our study, we already analyzed a published scRNA-seq dataset (Fig. 1c), which revealed a strong inverse relationship between GATA6 expression and EMT-associated programs. This supports the interpretation that GATA6 marks an epithelial-like tumor cell state.

We propose that GATA6 may reflect a dynamic transcriptional state rather than a fixed subtype identity, particularly given its variable expression levels across tumor cells. As a transcription factor with pleiotropic roles including in immunomodulation, we chose to focus on the functional implications of its expression pattern, especially in shaping the tumor–immune microenvironment, rather than fitting it into a rigid subtype classification. We hope this clarifies our rationale and approach.

More mouse cell lines needed for in vitro mechanism experiments

Response:

→ We agree with the reviewer, and in the revised manuscript, we have substantially increased the number of murine cell lines for *in vitro* experiments (Fig. 2,4). Besides, in addition to more *CKP*, *KPC* or *KPCY* derived cell lines, we included cell line models with GATA6 knock out and auxin-inducible degron (AID) GATA6 knock in (Fig. 2; Fig. S2) and additional 2 more cell lines with GP expression (Fig. 5).

The authors generated cell lines from syngeneic KPC mice. Since these lines were not subcloned, it is surprising that 3 of them show heterogeneous GATA6 expression, and 3 show no GATA6 expression at all (Figure 2A). The authors should genetically validate that these lines are exclusively tumor cells, and these experiments should be done with FACS-sorted GATA6-high and GATA6-low cells from several heterogeneous cell lines. The top histogram in Figure 2A shows an unusual bi-modal expression of GATA6 with many cells on the right y-axis. These data should be inspected for FACS artifacts, and a better example plot should be presented.

Response:

→ We thank the reviewer for this valuable comment and apologize for the lack of clarity in the original manuscript. The murine PDAC cell lines used in our study were derived from primary tumors of genetically engineered *CKP*, *KPC*, or *KPCY* mice, and each cell line was established from an independent tumor. While the lines were not subcloned, they were expanded from early-passage tumor cell outgrowths and have been validated to exhibit stable phenotypes.

We assessed GATA6 expression across 9 cell lines by flow cytometry and categorized them into GATA6^{high} (n=5) and GATA6^{low} (n=4) based on the median value of the mean fluorescence intensity. Importantly, each line exhibits a unimodal distribution of GATA6 signal, consistent with homogeneous expression within each culture. The previously noted “bi-modal” appearance in Figure 2a was due to the overlay of the GATA6 signal with the isotype control, rather than true subpopulations of differing GATA6 expression. We apologize for the lack of clarity in the original figure. To address this, we have revised Figure 2a to clearly label the isotype control and GATA6 antibody (GATA6 AB)-stained populations, which we believe will eliminate this confusion.

Additionally, it appears that only 4 of the 6 cell lines are used for the data in Figure 2c (it’s unclear how many were used for Figure 2b). How were these selected?

Response:

→ In Fig. 2b, we used 6 cell lines (three GATA6^{high}: 511950, 60400, 70301 vs three GATA6^{low}: 60531, 511892, 60590) for transcriptomic analysis. We acknowledge the limited representation in Fig. 2c, and in the revised manuscript, we have expanded the *in vitro* experiments to include all these 6 cell lines, as well as 2 additional cell lines to strengthen the conclusions (Fig. 2, Fig. 4). We have clarified this in the revised figure legends and Methods section.

The AID approach in Figure 2e-g more convincingly demonstrates the relationship between MEKi, GATA6, and antigen presentation, but should be performed on ≥3 distinct AID-altered lines.

Response:

→ Thank you for this constructive suggestion. To strengthen the mechanistic link between GATA6 and MEKi-induced anti-tumor responses, we performed additional *in vivo* experiments using the 110299^{AID-GATA6} cells (Fig. 3e-h; Fig. S3d-f). This inducible model allowed us to dynamically modulate GATA6 expression within the tumor microenvironment and evaluate its role in MEKi responsiveness.

Besides, in order to confirm the role of GATA6 in MEKi-induced MHC1 expression, we generated two CRISPR-Cas9 GATA6 knockout clones from the GATA6^{high} cell line 2838c3. In both clones, MEKi failed to induce MHC1 expression, confirming the essential role of GATA6 in this pathway (Fig. 2d-e; Fig. S2b). Together, these complementary models support the functional importance of GATA6 in mediating MEKi-induced tumor immunogenicity.

Additionally, more than 2 cell lines should be derived from the FKPC2GP mice to make conclusions that the distinct morphologies from the 2 cell lines match mechanisms related to GATA6 and MEK inhibition. Similarly, the data in Figure 4e appear to be from 2 KPC cell lines, this in vitro experiment should be performed on ≥5 cell lines with statistical comparisons shown.

Response:

→ In the revised manuscript, we have included 2 more GP-expressing cell lines for the co-culture experiment (Fig. 5). Additionally, for the *in vitro* combination treatment of MEKi and HDACi, we have substantially increased the number of murine PDAC cell lines to 8 (4 GATA6^{high} vs 4 GATA6^{low}) (Fig. 4).

The treatment regimen for the in vivo experiment in Supp Figure 3b is not clear. Did the recurrence occur during MEKi treatment or did MEKi withdrawal prompt the recurrence? Additionally, the authors should provide MR images from the whole cohort data.

Response:

→ We apologize for the unclear data presentation. After revision, the original Supplementary Figure 3 has now been reorganized and is now presented as the current Supplementary Figure 4 in the revised manuscript.

To address the reviewer's questions, we have now included a detailed treatment schedule and timeline in the revised figure (Fig. S4b). Importantly, MEKi treatment was administered continuously and not withdrawn; tumor recurrence occurred during ongoing MEKi therapy. Additionally, we have expanded the imaging data to improve transparency. The revised supplementary figure now includes MR images across the entire treatment period for the cohort (Fig. S4c). In addition to MR images (Fig. S4c), we have also included the images from ultrasound in Fig. 3c in order to illustrate the tumor volume changes over time. We hope these additions clarify the treatment regimen and imaging data.

The Figure 5e figure legend states that the MHC-I and GATA6 staining is from tumor cells but also from "whole tumorous tissue". This should be clarified. Figure 5g shows the percentage of tumor cells expressing GATA6 and MHC-I (though the y-axis is labelled only GATA6+) but it's unclear if the bars are superimposed or stacked and statistical comparisons are missing. Figure 6 shows that with the treatment effect (greatest in the MEKi+HDACi combination)

there is an increase in the percentage of leukocytes. Is this the percentage among all cells? As with earlier figures, these data would be easier to interpret with cell densities.

Response:

→ We apologize for the lack of clarity in the original figures and legends. After revision, the original Fig. 5 and Fig. 6 have now been reorganized and are now presented as the current Fig. 6 and Fig. 7 in the revised manuscript.

Regarding the term “tumorous tissue”, this refers to tumor regions delineated based on H&E and PanCK staining of the same tissue section, allowing us to exclude normal pancreatic areas from quantification. All analyses were restricted to these tumor-designated regions. This quantification approach was previously described in our publication in *Nature Communications*⁵ and utilizes both Definiens and HALO software.

For Fig. 5g (Fig. 6f in the revised manuscript), we have replotted the graph to show the individual values of GATA6⁺, GATA6⁺MHCI⁺, and GATA6⁺MHCI⁻ tumor cells, making the categories visually distinct. Statistical comparisons have now been included, and significance is indicated where applicable.

In the revised Fig. 7 (the original Fig. 6), all percentages refer to the proportion of marker-positive cells relative to the total number of nucleated cells within the tumor regions. We have clarified this in the figure legends to aid interpretation.

We hope these revisions clarify our quantification approach and improve data interpretation.

Additional Comments

Figure 1 and throughout, text on figure labels (especially axes labels) should be written in common syntax, without underscores and periods, and with appropriate superscripts.

Response:

→ We have revised the figure labels accordingly in the revised manuscript.

In Figure 1 (and throughout) for GSEA results, the authors should indicate which pathways are considered significant – the color and size scales for NES and FDR are difficult to interpret; nominal P values should also be given.

Response:

→ We have revised the figure labels accordingly in the revised manuscript. The pathways shown in the GSEA figures were all selected based on statistical significance (adjusted P-value < 0.05), as stated in the corresponding section of the text.

Numbers are missing from the y-axes of Figure 4b and Figure 4c making it difficult to compare data between the charts. The values for GATA6 expression are similar between the control conditions of GATA6-hi and GATA6-low tumors for Figure 4c. Did GATA6 expression change in vivo or are the y-axes different?

Response:

→ We apologize for the missing y-axis in Figure 4b and 4c, which may have been caused during image compression. We have included the y-axes of the graphs accordingly. Regarding the changes of GATA6 expression of the two *in vivo* models, the baseline levels and magnitude of changes are indeed different between the

GATA6^{high} and GATA6^{low} tumors, in which GATA6^{high} tumors show higher baseline of GATA6 expression, and there is a significant reduction of GATA6 expression upon MEKi treatment. On the contrary, GATA6 expression of GATA6^{low} tumors is low and there is no prominent change in GATA6 level upon MEKi.

Reviewer #4 (Remarks to the Author): with expertise in pancreatic cancer

Yang et al. investigated the impact of combined MEK inhibitor (MEKi) and histone deacetylase inhibitor (HDACi) treatment on GATA6-dependent MHC1 expression in pancreatic ductal adenocarcinoma (PDAC). The authors showed that MEKi alone increased MHC1 expression in GATA6-positive pancreatic cancer cells and triggers epithelial-to-mesenchymal transition that appeared to reduce the effect. The addition of HDACi (domatinostat), preserved the GATA6-positive population and enhanced MHC1 expression, resulting in improved immune response and decreased tumor growth. This combination therapy seemed to significantly increase the cytotoxic T-cell infiltration, enhancing anti-tumor responses. However, there appears to be a lack of coherence and consistency in linking the mechanistic insights across clinical data and experimental data. More importantly, it is not clear that MHC1 up-regulation is mechanistically important for the anti-tumor immunity upon the MEKi/HDACi combo treatment, while MEKi/HDACi combo treatment might have pleiotropic effects on both tumor and stroma. While the study presents a novel approach to enhancing immunogenicity in PDAC, the evidence supporting the main hypotheses needs to be significantly improved. The detailed points are listed below.

Major points

1. The clinical data analysis in Figure 1 shows an association between GATA6 expression and the inflammatory response signature including MHC1 up-regulation. However, it remains unclear how these findings relate to the therapeutic context of MEKi, since the patient dataset indicates these associations exist independent of MEK inhibition. The observation on clinical data is consistent with preclinical models? GATA6 positive cell lines tend to have the inflammatory response signature and MHC1 up-regulation (without MEKi)? As another example, in GATA6^{high} PDX vs. GATA6^{low} PDX in figure 4, do they have different MHC1 expression level?

Response:

→ We thank the reviewer for the important and insightful comments. The clinical data analysis in Fig. 1 highlights the immunological features of GATA6^{high} tumors and GATA6⁺ tumor cells, emphasizing their enhanced ability to interact with immune cells, particularly CD8⁺ T cells. These features exist independently of therapy, indicating that GATA6 expression is associated with a more immunogenic basal state.

In our preclinical models, we observed that GATA6^{high} tumor cells model (110299) tend to exhibit higher MHC1 expression even without treatment (Fig. 3d) compared to GATA6^{low} tumor cells model (60590, Fig. S3c), supporting the clinical observation. In murine PDAC cell lines, we also found GATA6^{high} lines tend to express higher levels of MHC1 at baseline (Fig. 4f). These findings support a conserved relationship between GATA6 expression and immunogenicity, consistent across human and mouse datasets (Fig. 4f). In the PDX models (Fig. 4), we did not observe a clear difference in

MHCI expression at baseline between GATA6^{high} and GATA6^{low} tumors. However, GATA6^{high} tumors consistently displayed greater MHC1 induction upon MEKi treatment. Importantly, our study focuses on the therapy-induced dynamics of these features. Specifically, while GATA6⁺ tumor cells respond to MEKi by upregulating MHC1 (as observed in the PDX model as well as CKP model, Fig. 4 and Fig. 6), GATA6 expression itself decreases following MEKi treatment (Fig. 4c, Fig. 6e, f), likely due to EMT induction. This therapy-induced mesenchymal shift limits sustained MHC1 expression, thereby constraining the immunogenic potential of the tumor. This dynamic regulation of GATA6 cannot be easily dissected in patient samples and requires controlled experimental model systems for detailed investigation.

Together, our findings support that GATA6 expression defines a tumor cell state that is both antigen-presenting and immune-permissive, and that MEKi transiently enhances this immunogenicity before inducing a GATA6^{low} mesenchymal phenotype. These insights provide the rationale for therapeutic strategies such as MEKi+HDACi to maintain GATA6 expression and prolong tumor sensitivity to immune attack.

2. The induction of MHC1 in GATA6+ pancreatic cancer cells upon MEKi treatment was shown in a couple of KPC cell lines. However, this does not appear robust enough or whether similar dynamics occur in human PDAC cell lines across different molecular subtypes (PDAC cell lines known to belong to classical subtype vs. basal-like).

Response:

→ Thank you for the constructive advice. In the revised version, we have expanded our cell line panel for the *in vitro* experiment. 8 murine PDAC cell lines (4 GATA6^{high} vs 4 GATA6^{low}) were included for the *in vitro* MEKi and/or HDACi experiments (Fig. 2c; Fig. 4d-f). This broader panel allows us to more robustly assess the association between GATA6 expression and MHC1 induction. To further validate the functional role of GATA6, we employed two complementary genetic approaches in GATA6^{high} murine cell lines: (1) an auxin-inducible degradation (AID) system in the 110299 cell line (Fig. 2f-h; Fig. S2c-g), and (2) CRISPR-Cas9 mediated knockout in the 2838c3 cell line (Fig. 2d-e; Fig. S2b). In both models, GATA6 loss abrogated MEKi-induced MHC1 upregulation, strengthening our conclusion that GATA6 is required for this effect.

→ Regarding human tumor cell responses, instead of using commercial human cell lines, we analyzed a panel of 15 patient-derived xenograft (PDX) models. In the revised manuscript, we now include the staining of KRT81, a basal-like subtype marker, to classify these models into classical or basal-like subtypes (Fig. S5b). This allows us to relate subtype identity to MHC1 responses upon MEKi treatment. Our findings show that MEKi-induced MHC1 upregulation occurs primarily in GATA6⁺ (classical) tumors, supporting the relevance of our observations in both mouse and human systems.

3. In figure 3, the analysis seems limited as it tests GATA6-dependent MHC1 expression and T-cell cytotoxicity mainly in one cell line each. Expanding this to more diverse cell lines could strengthen the conclusions drawn. It is not clear whether differential expression of MHC1 is functionally important to determine anti-tumor activity.

Response:

→ We thank the reviewer for this important suggestion. In response, the original Figure 3 has been reorganized and is now presented as Figure 5 in the revised manuscript.

To address the concern about limited cell line diversity, we expanded our co-culture experiments to include a total of four GP-expressing cell lines (2 GATA6^{high} vs. 2 GATA6^{low}). We observed that both GATA6^{high} GP-expressing cell lines respond to MEKi and/or HDACi by upregulating MHCI expression to a greater extent than the GATA6^{low} cell lines (Fig. 5d), and that this increase correlates with enhanced T cell-mediated anti-tumor cytotoxicity (Fig. 5e-f). We deduced that MHCI upregulation contributes at least in part to the observed increase in anti-tumor cytotoxicity.

To further support the association between MHCI and tumor cytotoxicity, we performed mIF analysis in the *CKP* mouse model treated with MEKi and/or HDACi (Fig. 7c-e). Our findings show that GzmB⁺ cells were more frequently found in close proximity to MHCI⁺ tumor cells, which were undergoing apoptosis (CI casp3⁺). This suggests that the enhanced cytotoxic cell-mediated cytotoxicity was specifically directed at MHCI⁺ tumor cells, further reinforcing the link between MHCI expression and anti-tumor activity.

To validate the role of GATA6 in the MEKi-induced MHCI and T cell cytotoxicity *in vivo*, we employed a targeted GATA6 protein degradation system using the orthotopic transplantation model of AID-GATA6-KI cells (Fig. 3e-h; Fig. S3d-f). With the AID-GATA6-KI system, we were able to reduce GATA6 protein expression after tumor establishment, and GATA6 levels remained low throughout the treatment period upon auxin administration (Fig. 3g; Fig. S3d). We found that GATA6 degradation significantly impaired MEKi-induced tumor suppression in the 110299^{AID-GATA6} orthotopic model (Fig. 3f). Additionally, the MEKi-induced upregulation of tumor MHCI expression, CD8 infiltration, and anti-tumor cytotoxicity were all markedly reduced (Fig. 3h), highlighting GATA6's essential role in MEKi-induced anti-tumor responses. Notably, MEKi-induced CD4 infiltration was unaffected by GATA6 degradation (Fig. S3f), suggesting that GATA6 specifically mediates the MEKi effect via the MHCI-CD8 pathway.

4. The introduction of HDAC inhibitors is abrupt and their role in the combinatory treatment effect alongside MEKi is not sufficiently explained.

Response:

→ In this study, HDAC inhibitors were chosen because HDACs play a key role in promoting EMT, and HDAC inhibition has been used in various cancer models to counteract EMT. In PDAC, Domatinostat has been shown to promote epithelial gene expression via a BRD4- and MYC-dependent mechanism³.

Since MEKi treatment consistently reduces GATA6 levels, probably via EMT induction (Fig. S5e), we hypothesized that HDACi could restore GATA6 expression. Indeed, we demonstrated that HDACi treatment largely restores GATA6 expression *in vitro* (Fig. 4d,e) and *in vivo* (Fig. 6e,f), significantly improving tumor control and mouse survival (Fig. 6b,c). Besides, we also demonstrated in the revised manuscript that HDACi, indeed, counteracts the EMT induced by MEKi (Fig. S5e).

We have clarified these findings and their rationale in the revised manuscript. Importantly, the combination of MEKi and HDACi was used as a proof-of-principle strategy to restore GATA6⁺ tumor cells population, thereby enhancing treatment responsiveness. This supports the concept that targeting tumor cell plasticity can improve therapeutic outcomes.

5. When combo-treatment is done in PDX models (Figure 4), is the therapeutic effect mainly through MHCI-T cell mediated cytotoxicity? If so, PDX models in

immune-compromised or -deficient hosts show no obvious difference since they lack adaptive immune system?

Response:

→ The purpose of the PDX model is to demonstrate that, in human tumor cells, MEKi-induced MHCII upregulation occurs specifically in GATA6⁺ tumor cells, as observed in the murine cell line models. In the PDX model, MEKi treatment did not result in significant tumor suppression. Since the PDX model consists of immunodeficient nude mice that lack functional T cells, the observation of higher MHCII expression on GATA6⁺ cells without a corresponding increase in apoptotic cells suggests that T cells are essential for MHCII-mediated cytotoxicity.

6. Figure 5e data seems contradictory with what the authors showed so far. MEKi treatment did not up-regulate MHCII expression. HDACi treatment didn't seem to retain GATA6 positive population either. Only combo context resulted in up-regulation of MHCII (no change in GATA6 positive population). This also raise a concern whether the main hypothesis was thoroughly tested and the conclusion is supported by robust data.

Response:

→ The original Figure 5 has now been reorganized and is presented as the revised Figure 6. In the experiment originally shown in Fig. 5e (the revised Fig. 6f), we did not observe an increase in MHCII expression with MEKi treatment alone. This outcome is consistent with our mechanistic model: while MEKi upregulates MHCII expression in GATA6⁺ tumor cells, it simultaneously reduces the proportion of GATA6⁺ cells, likely through induction of epithelial-to-mesenchymal transition (EMT), as demonstrated in Fig. 4e and Fig. S5e. As a result, the overall MHCII signal remains unchanged or even reduced at the population level.

In contrast, HDACi treatment counteracts MEKi-induced EMT and thereby helps preserve the GATA6⁺ tumor cell population (Fig. 4e). Although HDACi alone does not significantly increase MHCII expression, it restores a GATA6⁺ epithelial cell state, which is more responsive to MEKi-induced MHCII upregulation. This mechanistic rationale led us to hypothesize that the combination of MEKi and HDACi would synergistically enhance MHCII expression.

Indeed, this hypothesis is supported by our data: combination treatment significantly increased MHCII expression and enhanced anti-tumor cytotoxicity across multiple models (Fig. 4, 5, 6, and 7). We have also revised the schematic diagram in Figure 8 to better illustrate the proposed mechanism, integrating the dynamic regulation of GATA6 and MHCII in response to therapy.

Taken together, our data robustly support the conclusion that GATA6 defines a tumor cell state that is both MHCII^{high} and immune-permissive, and that MEKi+HDACi combination therapy maximizes MHCII expression by preserving GATA6⁺ cells, while enhancing their immunogenicity.

Minor points

1. In Figure 3a, the figure is not complete.

Response:

→ We apologize for the incomplete figure, which was likely due the file compression during the submission process. We have carefully reviewed and corrected the figure in the revised manuscript, and we confirm that the complete version is now included.

2. In figure 4, the graphs do not have some information such as y-axis titles and units.

Response:

→ We also apologize for the missing axis titles and units in Figure 4, which again may have resulted from formatting or compression issues. In the revised manuscript, we have ensured that all graphs in Figure 4 now include complete axis labels and units for clarity and accurate interpretation.

3. In Figure 4, for the PDX experiments, the authors used mocetinostat as HDACi, and then switched to dometinostat. Any reason?

Response:

→ Domatinostat was used for the subsequent experiments because it shows greatest magnitude of MHC1 upregulation upon combination with MEKi in the *in vitro* experiment (Fig. S5f-g).

4. Figure 5e & 5g, it is questionable whether it accurately counted GATA6+ and MHC1+ cells. The positive cell population ranges only 2-8% in the entire population in a given area?

Response:

→ The original Figure 5 has now been reorganized and is presented as the revised Figure 6. We have carefully checked our quantification methods and confirm that the reported positive cell populations are accurate. The percentages refer to the proportion of cells positive for GATA6 or MHC1 out of the total number of cells present in the entire tumor tissue section. Given the heavy stromal infiltration in PDAC, it is common for stromal cells (including immune cells and fibroblasts) to constitute the majority of the tumor bulk, ranging from 50% to 80%, leaving only about 20% as tumor cells. Among these, GATA6⁺ tumor cells account for 2% to 8% of the tumor cell population, which is consistent with previous findings in PDAC patient cohorts, where the GATA6⁺ subtype represents only a subset of PDAC tumors.

To improve clarity, we have revised the corresponding figure legends in the revised manuscript (Fig. 6e, f) explicitly define the “positive percentage” as the percentage of positive cells among all cells in the tumor tissue.

References

1. Zhao B, *et al.* PLK1 blockade enhances the anti-tumor effect of MAPK inhibition in pancreatic ductal adenocarcinoma. *Cell reports* **44**, (2025).
2. Subramanian S, Bates SE, Wright JJ, Espinoza-Delgado I, Piekarz RL. Clinical toxicities of histone deacetylase inhibitors. *Pharmaceuticals* **3**, 2751-2767 (2010).
3. Mishra VK, *et al.* Histone deacetylase class-I inhibition promotes epithelial gene expression in pancreatic cancer cells in a BRD4-and MYC-dependent manner. *Nucleic Acids Res* **45**, 6334-6349 (2017).

4. De Sanctis F, *et al.* Expression of the membrane tetraspanin claudin 18 on cancer cells promotes T lymphocyte infiltration and antitumor immunity in pancreatic cancer. *Immunity* **57**, 1378-1393. e1314 (2024).
5. Cheung PF YJ, Fang R, Borgers A, Krenzel K, Stoffel A, *et al.* Progranulin mediates immune evasion of pancreatic ductal adenocarcinoma through regulation of MHCI expression. *Nat Commun* **13**, (2022).

REVIEWERS' COMMENTS

Reviewer #1 (Remarks to the Author):

The authors have sufficiently addressed my original concerns.

One additional concern is over Reference 35, which is bioRxiv from 2022. It is unclear whether Nature allows non-peer review references. Additionally, the age of this reference is also concerning, because it has not been officially published for over 3 years. I would recommend that essential data be added as supplemental for this manuscript or anything associated with that reference be removed as it has not been adequately peer reviewed.

Thank you for your comment and suggestion. We have now replaced Reference 35 with a recently peer-reviewed and published article:

Godfrey LK, et al. *Pancreatic cancer acquires resistance to MAPK pathway inhibition by clonal expansion and adaptive DNA hypermethylation*. Clin Epigenet 16, 13 (2024).

Reviewer #2 (Remarks to the Author):

The authors exhaustively addressed all critical comments from my and other reviewers. I believe the manuscripts is substantially improved and can be an exciting contribution to the field.

We sincerely thank the reviewer for the positive comments.

Reviewer #3 (Remarks to the Author):

The authors have done a nice job addressing the concerns from the initial review. They've clarified the writing, added important new experiments (including the GATA6 knockout/AID models and CD8 depletion studies), and expanded both the cell line and patient tumor data to strengthen the conclusions. The rationale for including HDAC inhibition is now much clearer, and they've been transparent about its limitations.

We sincerely appreciate the reviewer's positive feedback.

Reviewer #4 (Remarks to the Author):

In the revised manuscript, the authors have provided more robust data to support GATA6-dependent MHCI expression in the context of MEK inhibition. The new data clearly strengthen the conclusion of the revised manuscript. Although the authors effectively showed that GATA6 expression is associated with a more immunogenic basal state, GATA6 does not seem to be responsible for up-regulation of MHCI expression in the basal state as shown in Figure 2d-h. GATA6 knock-outs or degradation have no effect on the basal level of MHCI, while GATA6-high vs. low cells showed a trend of MHCI expression difference. This difference should be clearly documented and acknowledged.

The authors have addressed all other concerns in the revised manuscript.

We thank the reviewer for the thorough evaluation of our data and the positive comments.